# CRYOLVM: SELF-SUPERVISED LEARNING FROM CRYO-EM DENSITY MAPS WITH LARGE VISION MODELS

**Weining Fu[1,†]**  **Kai Shu[2,†]**  **Kui Xu[3,\*]**  **Qiangfeng Cliff Zhang[1,\*]**

[1] State Key Laboratory of Membrane Biology, Beijing Frontier Research Center of Biological Structures, Tsinghua-Peking Joint Center for Life Sciences, School of Life Sciences, Tsinghua University, Beijing, China
[2] State Key Laboratory of Membrane Biology-Membrane Structure and Artificial Intelligence Biology Branch, Hangzhou, China
[3] State Key Laboratory of Membrane Biology, Beijing Tsinghua Institute for Frontier Interdisciplinary Innovation, Beijing, China

[†] These authors contributed equally to this work.
[\*] Correspondence: `xukui.2016@tsinghua.org.cn`, `qczhang@tsinghua.edu.cn`

## ABSTRACT

Cryo-electron microscopy (cryo-EM) has revolutionized structural biology by enabling near-atomic-level visualization of biomolecular assemblies. However, the exponential growth in cryo-EM data throughput and complexity, coupled with diverse downstream analytical tasks, necessitates unified computational frameworks that transcend current task-specific deep learning approaches with limited scalability and generalizability. We present CryoLVM, a foundation model that learns rich structural representations from experimental density maps with resolved structures by leveraging the Joint-Embedding Predictive Architecture (JEPA) integrated with a SCUNet-based backbone, enabling rapid adaptation to various downstream tasks. We further introduce a novel histogram-based distribution alignment loss that accelerates convergence and enhances fine-tuning performance. Through comprehensive evaluation across three critical cryo-EM tasks—density map sharpening, density map super-resolution, and missing wedge restoration—CryoLVM consistently outperforms state-of-the-art baselines across multiple density map quality metrics, confirming its potential as a versatile foundation model for a wide spectrum of cryo-EM applications.

## 1 INTRODUCTION

All biological functions are governed by events orchestrated at the molecular level by macromolecular assemblies (Liao et al., 2013; Gestaut et al., 2022). Elucidating their intricate structures not only deepens our understanding of underlying molecular mechanisms, but also lays the groundwork for advancements in fields like drug discovery (Congreve et al., 2020). Cryo-electron microscopy has emerged as a pivotal technique for resolving macromolecular structures (Kühlbrandt, 2014; Nogales & Mahamid, 2024), with deposited density maps in the Electron Microscopy Data Bank (EMDB) (wwPDB Consortium, 2023) growing exponentially (Lawson et al., 2016). However, cryo-EM faces several intrinsic pitfalls that complicate accurate structure determination. Single particle analysis (SPA) cryo-EM suffers from poor signal-to-noise ratio, attenuated high-frequency information, and anisotropic resolution arising from low-dose imaging requirements, sample heterogeneity, and preferred specimen orientation(Rosenthal & Henderson, 2003; Liu et al., 2025). Cryo-electron tomography (cryo-ET) encounters additional limitations from limited per-image electron dose, missing wedge artifacts, and substantial noise that obscure fine structural details (Liu et al., 2022a; Tao et al., 2018). These challenges hinder direct structural interpretation from raw cryo-EM maps, particularly at resolutions lower than 4 Å, necessitating extensive processing pipelines with substantial manual intervention and domain expertise (Terwilliger et al., 2018a; Terashi & Kihara, 2018).

Building on these observations, the cryo-EM community has increasingly embraced machine learning approaches across the entire structural determination pipeline. For map reconstruction, cryo-DRGN (Zhong et al., 2021) employs variational autoencoders to learn continuous latent representations of conformational heterogeneity, while 3DFlex (Punjani & Fleet, 2023) models non-rigid motions through coordinate-based neural networks. For postprocessing, DeepEMhancer (Sanchez-Garcia et al., 2021) adopts a 3D U-Net architecture trained on pairs of experimental maps and LocScale-sharpened targets for automated masking and local sharpening, while EMReady (He et al., 2023) implements a Swin-Conv-UNet framework combining residual convolution for local modeling with Swin transformer for non-local feature extraction. For atomic model building, deep learning approaches have progressed from early methods such as A2-Net (Xu et al., 2019) and DeepTracer (Pfab et al., 2021), which perform de novo amino acid recognition and backbone tracing from density maps, to more recent advances like ModelAngelo (Jamali et al., 2024), which employs graph neural networks with iterative refinement and leverages protein language model embeddings to jointly perform atomic coordinate prediction and sequence assignment, achieving automated model building at expert-level quality. Despite these advances, current deep learning methods in cryo-EM remain predominantly task-specific and rely on supervised training paradigms requiring labeled input data, constraining dataset scales and yielding models with limited generalizability.

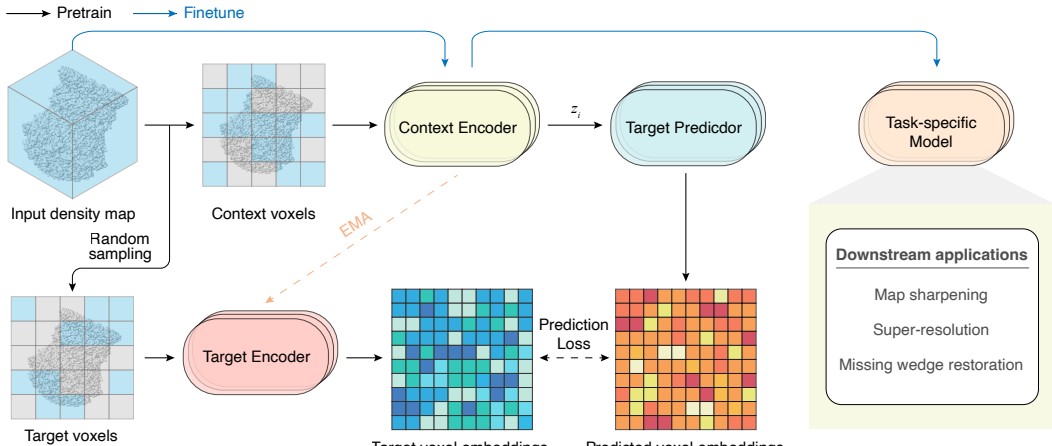

Figure 1: CryoLVM framework. During pretraining, input density maps are partitioned into non-overlapping 3D patches, which are then randomly divided into context and target subsets. The Context Encoder processes visible context patches, and the Target Predictor takes the resulting context embeddings along with positional information of masked target patches to predict the corresponding Target Encoder outputs. A regression loss is applied between predicted and target voxel embeddings to encourage representational alignment. The weights of the Target Encoder are updated via an exponential moving average (EMA) of the Context Encoder weights. Following pretraining, the pretrained encoder is adapted to three downstream cryo-EM tasks through fine-tuning with task-specific models: density map sharpening, super-resolution, and missing wedge restoration.

Foundation models harness self-supervised pretraining on large-scale data to develop transferable representations, facilitating efficient adaptation to diverse downstream applications (Wei et al., 2022; Achiam et al., 2023; Kaplan et al., 2020). Notably, protein language models trained on extensive sequence corpora have become transformative tools for structure prediction (Lin et al., 2023), protein design (Verkuil et al., 2022), and function annotation (Hayes et al., 2025). While foundation models have recently emerged for 2D cryo-EM image processing tasks (Shen et al., 2024; Zhang et al., 2025; Yan et al., 2024), they remain largely unexplored in the cryo-EM density map domain. Zhou et al. (2024) introduced CryoFM, a flow-based foundation model for cryo-EM density maps that demonstrates versatility as a generative prior across multiple tasks. Nevertheless, CryoFM was trained and evaluated exclusively on curated high-quality density maps, with assessments conducted primarily on synthetic noise-corrupted maps. Consequently, its robustness and performance on genuinely noisy, low-resolution experimental maps from real cryo-EM workflows remain to be established.

We present CryoLVM, a foundation model that introduces Joint-Embedding Predictive Architecture (JEPA) (Assran et al., 2023) to the cryo-EM density map domain, learning self-supervised struc-

tural representations that facilitate efficient adaptation across multiple downstream tasks. Unlike reconstruction-based pretraining methods such as masked autoencoders that predict raw voxel values, JEPA operates in abstract representation space, naturally filtering the pervasive noise in cryo-EM densities while capturing high-level structural semantics. We employ a SCUNet-based encoder (Zhang et al., 2023) within the JEPA framework as the backbone architecture, whose hybrid design combining Swin Transformer and residual convolution pathways is well-suited for cryo-EM data, where both local atomic-level features and global cross-regional spatial relationships must be jointly modeled. Through pretraining on experimental cryo-EM density maps with resolved structures collected from the EMDB, followed by fine-tuning with task-specific decoders, CryoLVM attains state-of-the-art results across three critical downstream tasks — density map sharpening, super-resolution, and missing wedge restoration — demonstrating robust performance on genuinely noisy experimental maps and its promise as a versatile foundation model for broader cryo-EM applications.

The main contributions of this work are summarized as follows:

- We propose CryoLVM, the first foundation model to employ JEPA with a SCUNet-based backbone for self-supervised learning on 3D cryo-EM density maps, effectively combining local atomic-level feature extraction with global spatial context modeling to learn semantically rich structural representations;

- We develop a novel histogram-based distribution alignment loss $\mathcal{L}_{\text{HistKL}}$ that, when combined with standard reconstruction loss, enhances convergence speed and improves fine-tuning performance;

- We conduct comprehensive experiments across three downstream tasks, demonstrating that CryoLVM consistently outperforms existing methods including DeepEMhancer, EMReady, EM-GAN (Subramaniya et al., 2021), and IsoNet (Liu et al., 2022a) on most evaluation metrics.

## 2 RELATED WORK

**Density map sharpening** Cryo-EM density maps suffer from resolution-dependent amplitude falloff that attenuates high-frequency contrast, necessitating post-processing methods to restore interpretability. Traditional map sharpening approaches include global methods like phenix.auto_sharpen (Terwilliger et al., 2018b) and RELION (Scheres, 2015) that apply uniform B-factor correction, and local methods such as LocScale (Jakobi et al., 2017) that perform spatially adaptive enhancement by comparing the local amplitude spectrum of the experimental map with that of a reference atomic model. Recent deep learning approaches have advanced this field significantly: DeepEMhancer employs a 3D U-Net architecture to mimic LocScale's local sharpening effects (Sanchez-Garcia et al., 2021), while EMReady utilizes a 3D Swin-Conv-UNet framework with combined smooth L1 and structural similarity losses to optimize both local and non-local features of experimental cryo-EM density maps (He et al., 2023).

**Density map super-resolution** Model building from low-resolution cryo-EM maps remains a major impediment, as current state-of-the-art methods like ModelAngelo exhibit marked performance degradation beyond 4 Å resolution (Jamali et al., 2024). For protein identification methods like CryoDomain (Dai et al., 2025), their accuracy also deteriorates as resolution exceeds 6-8 Å. These limitations motivate the development of super-resolution techniques, which aim to estimate high-resolution maps from their low-resolution counterparts and thereby facilitate downstream structural determination. EM-GAN represents an early deep learning approach in this domain, employing 3D generative adversarial networks to enhance experimental cryo-EM maps in the 3-6 Å resolution range (Subramaniya et al., 2021).

**Missing wedge restoration in cryo-ET** The missing wedge problem in cryo-ET stems from physical limitations during tilt-series acquisition, where specimen geometry and mechanical constraints restrict image collection to an angular range of typically $\pm 60°$ (Lučić et al., 2005). Incomplete angular sampling creates characteristic wedge-shaped gaps in Fourier space, leading to anisotropic resolution with severe artifacts along the beam direction, manifesting as structural elongation and distortion that compromise reconstruction fidelity (Wiedemann & Heckel, 2024). While classical approaches have implemented iterative reconstruction algorithms including SIRT (Gilbert, 1972) and ART (Gordon et al., 1970; Yan et al., 2019), alongside constrained optimization techniques like ICON (Deng et al., 2016) that impose binary assumptions such as density positivity, these methods offer limited recovery of missing information. Deep learning has revolutionized miss-

ing wedge restoration through learning complex priors directly from tomographic data. IsoNet, built upon a 3D U-Net architecture, iteratively recovers missing information by training on paired datasets—generated by rotating subtomograms to 20 orientations and imposing additional missing wedge artifacts—to map degraded inputs to less-degraded targets (Liu et al., 2022a).

# 3 METHODOLOGY

## 3.1 MODEL ARCHITECTURE

CryoLVM combines Joint-Embedding Predictive Architecture (JEPA) with SCUNet backbone for efficient self-supervised learning on cryo-EM density maps. Our design addresses the unique challenges of volumetric biological data through strategic architectural modifications and domain-specific optimizations.

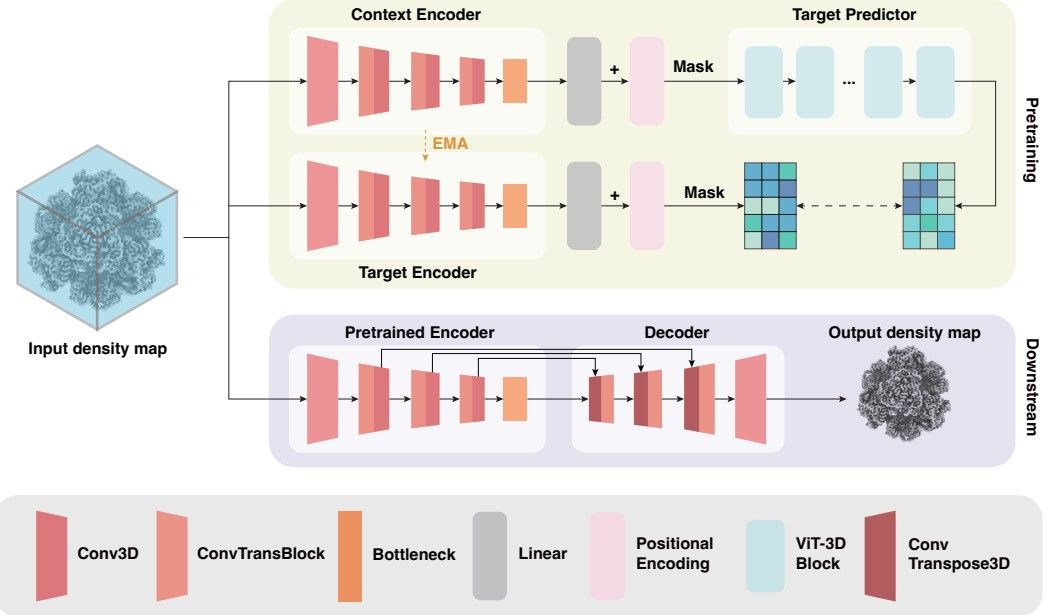

Figure 2: CryoLVM architecture. The Context Encoder and Target Encoder use hierarchical swin-conv (SC) blocks for multi-scale feature extraction, with outputs converted to patch embeddings and combined with 3D sinusoidal positional encodings. The Target Predictor employs transformer blocks to predict target representations. For downstream tasks, task-specific decoder with upsampling SC blocks are jointly fine-tuned with the pretrained encoder.

We adopt JEPA over traditional masked autoencoders for its representation-space prediction paradigm. Unlike voxel-level reconstruction that amplifies noise in low-SNR cryo-EM densities, JEPA learns semantic features by predicting masked region representation from visible context. This approach naturally filters noise while preserving structural information critical for downstream tasks. Our pretraining architecture comprises three components: Context Encoder $f_{\theta_c}$, Target Encoder $f_{\bar{\theta}_t}$ (stop-gradient), and Target Predictor $g_\phi$. The training objective minimizes prediction loss:

$$\mathcal{L}_{\mathrm{p}} = \mathbb{E}_{x,M}[\Sigma_{i \in M}\mathrm{SmoothL1}_\beta(g_\phi(f_{\theta_c}(x_{\mathrm{context}}), z_i) - f_{\bar{\theta}_t}(x_i))], \tag{1}$$

where $M$ denotes masked patches, $z_i$ denotes spatial position information, and $\mathrm{SmoothL1}_\beta$ provides robust optimization with reduced sensitivity to outliers compared to L2 loss.

Different from previous studies, we select SCUNet over conventional 3D Vision Transformer as the backbone for its hybrid architecture that adeptly combines local and global modeling. SCUNet's swin-conv (SC) blocks partition features into dual paths: Swin Transformers capture long-range dependencies while residual convolutions preserve local details. This architectural choice is particularly well-suited for cryo-EM applications, where local convolutions extract atomic-level structural

features and global attention mechanisms model cross-regional spatial relationships across multiple scales. Our Context Encoder and Target Encoder consist of three downsampling SC blocks, three 3D convolution blocks and a bottleneck SC block. Within each SC block, input features are bifurcated into parallel pathways: the convolution branch employs residual $3 \times 3 \times 3$ convolutions with Filter Response Normalization, while the transformer branch utilizes 3D windowed multi-head self-attention with window size $4 \times 4 \times 4$. Feature fusion is achieved through $1 \times 1 \times 1$ convolutions, enabling simultaneous local feature preservation and global context aggregation. During pretraining, encoder outputs are converted to patch embeddings via linear transformation and incorporated with 3D sinusoidal positional encodings to preserve spatial relationships within the feature grid. Masks are then applied to these patch embeddings to generate context and target embeddings. The Target Predictor follows the JEPA framework, comprising standard transformer blocks. Final predictions are mapped back to encoder embedding dimension through a linear projection, ensuring compatibility with target outputs for loss computation. For downstream applications, task-specific decoders utilize upsampling SC blocks and 3D transposed convolution blocks, jointly fine-tuned with the pretrained encoder on labeled datasets for each target task.

## 3.2 Loss For Downstream Tasks

Across all three downstream tasks, we employ a unified composite loss function that combines standard reconstruction error $\mathcal{L}_{\text{MSE}}$ with distributional alignment loss $\mathcal{L}_{\text{HistKL}}$.

To enforce statistical consistency between predicted and target densities, we design a novel histogram-based distribution alignment loss, denoted as $\mathcal{L}_{\text{HistKL}}$. The idea is to align the predicted density distribution with the ground-truth density distribution via a differentiable histogram and a divergence measure. Given predicted density $X$ and target density $X^{\star}$, we first construct their soft histograms using Gaussian kernel weighting:

$$h(x)_j = \frac{1}{N} \sum_{i}^{N} \exp\left(-\frac{1}{2}\left(\frac{x_i - c_j}{\sigma}\right)^2\right),$$ (2)

where $c_j$ denotes the center of the $j$-th bin, $\sigma$ controls the smoothness, and $N$ is the number of total voxels.

Then, we quantify the distributional divergence between the two histograms $p = h(X)$ and $q = h(X^{\star})$ using Jensen–Shannon (JS) divergence, which is based on Kullback-Leibler (KL) divergence:

$$D_{\text{JS}}(p \parallel q) = \tfrac{1}{2} \sum_{k} p_k \log \frac{p_k}{m_k} + \tfrac{1}{2} \sum_{k} q_k \log \frac{q_k}{m_k},$$ (3)

with $m = \frac{1}{2}(p + q)$.

The proposed histogram KL loss is thus defined as:

$$\mathcal{L}_{\text{HistKL}}(X, X^{\star}) = D_{\text{JS}}\big(H(X) \,\big\|\, H(X^{\star})\big).$$ (4)

The final objective combines both loss components:

$$\mathcal{L}_{\text{total}} = \alpha \, \mathcal{L}_{\text{MSE}} + (1 - \alpha) \, \mathcal{L}_{\text{HistKL}},$$ (5)

where hyperparameter $\alpha \in [0, 1]$ balances reconstruction accuracy and distributional alignment.

# 4 EXPERIMENTS

## 4.1 Pretraining Dataset

To enable effective learning of structural semantics embedded within cryo-EM density maps, we assembled our pretraining dataset by leveraging the training subset of Cryo2StructData (Giri et al., 2024), currently the most comprehensive publicly accessible repository in this domain. This curated collection comprises 7,392 high-resolution experimental density maps representing proteins and macromolecular complexes with resolutions spanning 1-4 Å. All density maps underwent standardized preprocessing procedures: 1) voxel sizes were uniformly resampled to 1 Å spacing, 2) density

values were clipped to the 0.01-0.99 percentile range to mitigate outlier effects and subsequently normalized to $[0, 1]$. To ensure rigorous evaluation and prevent data leakage, we systematically excluded density maps present in the test sets of baseline methods used for downstream task comparisons, yielding a final pretraining corpus of 7,302 density maps. For CryoLVM input generation, we applied random spatial cropping to extract volumes of size $48^3$.

Table 1: Evalutation metrics of different methods on the density map sharpening test set.

| | phenix.map_model_cc | | | Chimera.MapQ |
|---|---|---|---|---|
| | $CC_{box} \uparrow$ | $CC_{mask} \uparrow$ | $CC_{peaks} \uparrow$ | Q-score $\uparrow$ |
| Deposited | 0.744 | 0.788 | 0.659 | 0.338 |
| DeepEMhancer | 0.695 | 0.679 | 0.659 | 0.323 |
| EMReady | 0.878 | 0.802 | 0.791 | 0.424 |
| CryoLVM | **0.894** | **0.821** | **0.806** | **0.444** |

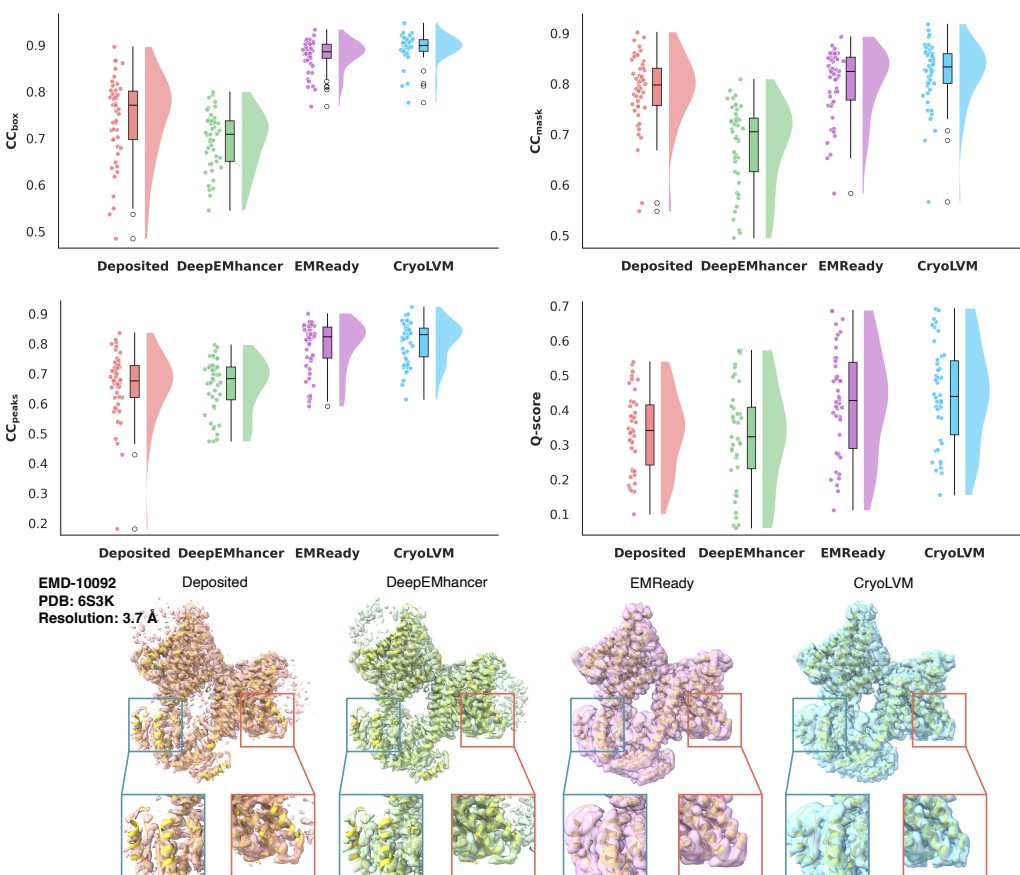

Figure 3: Comparative evaluation of density map sharpening performance across baseline and proposed methods. Cross-correlation metrics ($CC_{box}$, $CC_{mask}$, $CC_{peaks}$) were calculated via phenix.map_model_cc (Afonine et al., 2018), which quantify the agreement between density maps and their associated atomic models over different spatial regions. Q-score was computed using Chimera.MapQ (Pintilie et al., 2020; 2025), providing an independent assessment of map quality based on local atom-to-density correlation and atomic resolvability.

## 4.2 DENSITY MAP SHARPENING

Cryo-EM density maps suffer from resolution-dependent amplitude falloff that attenuates high-frequency contrast, making direct structural interpretation challenging. Density map sharpening

aims to computationally recover high-frequency information while preserving structural integrity, effectively reversing the B-factor decay that obscures fine molecular details.

Following the established training paradigm of EMReady, we adopted a supervised learning approach for density map sharpening using paired experimental and target maps. Target density maps were simulated from their corresponding structures using Chimera.molmap (Pettersen et al., 2004), with the resolution parameter matched to that of the paired experimental maps.

We evaluated CryoLVM on density map sharpening using a comprehensive set of quality metrics computed through established cryo-EM analysis tools. As shown in Tab. 1, CryoLVM demonstrates superior performance across all metrics, achieving the highest scores in $CC_{box}$ (0.894), $CC_{mask}$ (0.821), $CC_{peaks}$ (0.806), and Q-score (0.444). These improvements indicate enhanced overall quality between sharpened maps, suggesting better preservation of structural features and more accurate high frequency signal recovery. The violin plots in Fig. 3 also illustrate that CryoLVM consistently achieves tighter distributions with higher median values, indicating robust performance across diverse map types and resolution ranges. Visualization for EMD-10092 in Fig. 3 showcases CryoLVM's ability to enhance fine structural details such as improved definition of secondary structure elements while maintaining overall molecular topology.

Table 2: Evalutation metrics of different methods on the density map super-resolution test set

|  | phenix.mtriage | | | CryoRes |
| --- | --- | --- | --- | --- |
|  | $d_{model}(\text{Å})\downarrow$ | FSC-0.143(Å) $\downarrow$ | FSC-0.5(Å) $\downarrow$ | Resolution(Å) $\downarrow$ |
| Deposited | 3.66 | 3.39 | 4.64 | 3.81 |
| DeepEMhancer | 3.27 | 2.72 | 4.86 | 3.47 |
| EMGAN | 2.49 | 2.70 | 5.46 | 4.18 |
| CryoLVM | **2.33** | **2.58** | **4.58** | **3.39** |

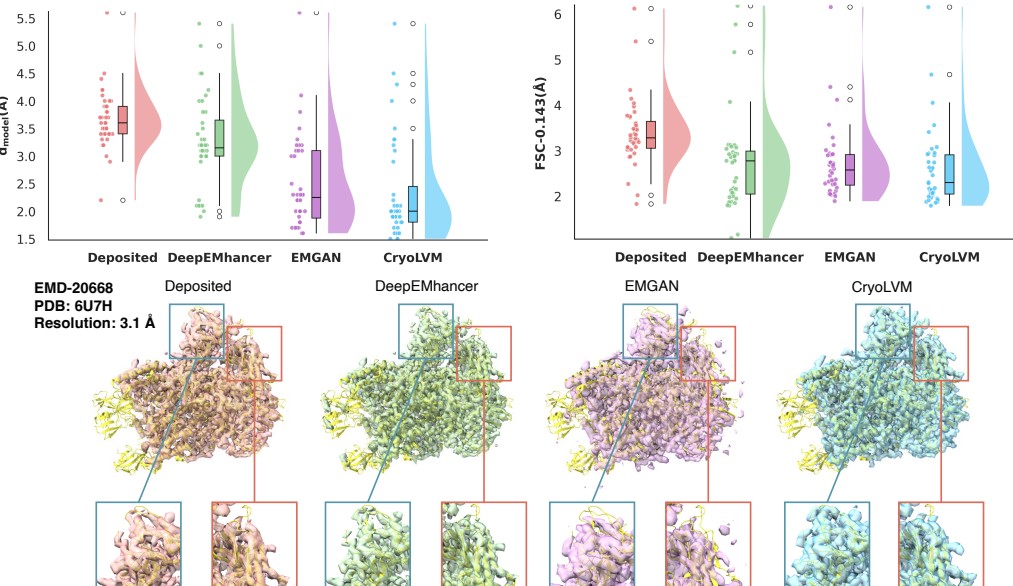

Figure 4: Comparison of map super-resolution performance between different methods. FSC-based metrics computed using phenix.mtriage (Afonine et al., 2018). Local resolution estimates obtained via CryoRes (Dai et al., 2023); global resolution represents the average of voxel-wise predictions.

## 4.3 DENSITY MAP SUPER-RESOLUTION

Cryo-electron microscopy (cryo-EM) frequently produces density maps at intermediate resolutions that preclude direct atomic model building. Even modest resolution improvements can greatly ben-

efit downstream structural analysis. Fine-grained density features are indispensible for accurately determining backbone geometry and side-chain conformations.

We developed a supervised training protocol for CryoLVM based on paired density maps. Training data consists of experimental maps at 4-6 Å paired with target maps simulated from their corresponding PDB structures. Target maps were generated using Chimera.molmap with the resolution parameter set to 1.8 Å. This paired supervision constrains the model to learn a robust mapping from medium-resolution experimental densities to near-atomic structural detail, enabling effective super-resolution of cryo-EM maps.

We benchmarked CryoLVM for density map super-resolution against DeepEMhancer and EMGAN. Under the phenix.mtriage (Afonine et al., 2018) metrics (Fig. 4), CryoLVM achieves the best FSC-based resolutions ($d_{model}$=2.33Å, FSC-0.143 = 2.58, FSC-0.5 = 4.58), significantly outperforming both DeepEMhancer and EMGAN. Similarly, when assessed with CryoRes (Dai et al., 2023), CryoLVM attains 3.39Å versus 3.47Å and 4.18Å for the baselines. Visualizations for EMD-20668 (PDB: 6U7H, 3.1 A) are shown in Fig. 4 . In the two magnified insets, CryoLVM improves backbone connectivity, and aligns much more closely with ground truth. CryoLVM not only enhances better visual interpretability but also improves quantitative resolution metrics, facilitating more accurate downstream structure determination from intermediate-resolution cryo-EM maps.

### 4.4 MISSING WEDGE RESTORATION

Cryo-electron tomography (cryo-ET) faces critical technical constraints: radiation sensitivity limits total electron dose, resulting in low signal-to-noise ratios, while specimen geometry restricts tilt angles to approximately $\pm 60°$ rather than the ideal $\pm 90°$. As shown in Fig. 5, this limited angular sampling creates characteristic wedge-shaped gaps in Fourier space. Missing wedge produces anisotropic resolution with degraded Z-axis information and structural distortions. We simulated missing wedge artifacts by applying a Fourier transform to the map and then masking the frequency domain with a wedge-shaped filter.

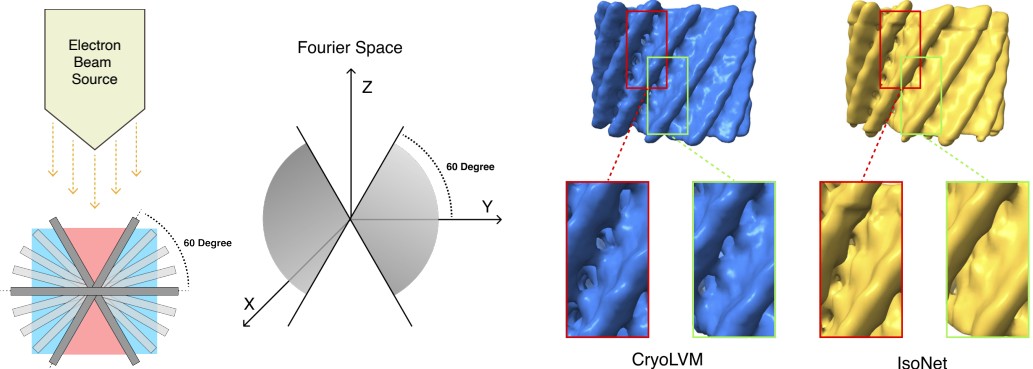

Figure 5: Visualization and schematic analysis of the missing wedge problem and its restoration. Left: schematic illustration of the missing wedge problem in cryo-electron tomography. Right: case result for missing wedge task. The original density map is EMD-5331 from EMDB and is added wedge-shaped mask for model processing.

**Fourier-domain wedge masking.** Given a 3D volume $V$, let its Fourier transform be $\widehat{V}(\mathbf{k})$ at frequency $\mathbf{k} \in \mathbb{R}^3$. To simulate the missing wedge, we introduce an indicator mask $M(\mathbf{k})$ that preserves only frequencies within the permitted tilt cone:

$$M(\mathbf{k}) = \begin{cases} 1, & |\arctan(k_z/\sqrt{k_x^2 + k_y^2})| \leq \theta_{\max}, \\ 0, & \text{otherwise.} \end{cases}$$

The corrupted spectrum is then

$$\widetilde{V}(\mathbf{k}) = M(\mathbf{k})\,\widehat{V}(\mathbf{k}),$$

and the simulated tomogram with missing wedge effect is obtained by inverse Fourier transform,

$$V_{\mathrm{mw}}(\mathbf{x}) = \mathcal{F}^{-1}\{\widetilde{V}(\mathbf{k})\}.$$

Here, $\theta_{\max}$ denotes the experimental tilt limit (commonly around $60°$).

We evaluated restoration performance via phenix.mtriage (Afonine et al., 2018) and report the map-model FSC-based resolution metrics $d_{\min}$ (Å) at two cutoff thresholds, FSC = 0.143 and FSC = 0.5. Lower $d_{\min}$ values indicate higher resolution and thus better restoration quality. Our results demonstrate that CryoLVM consistently outperforms IsoNet baseline across all evaluation metrics(Tab. 3): at FSC = 0.143, CryoLVM reduces $d_{\min}$ from 10.448 Å to 10.094 Å ($\sim 3.39\%$); at FSC = 0.5, CryoLVM improves from 12.361 Å to 11.447 Å ($\sim 7.39\%$). These quantitative improvements in $d_{\min}$ reflect CryoLVM's greater ability to recover high-frequency structural information that is typically lost due to missing wedge artifacts.

Fig. 5 depicts a missing-wedge case, where CryoLVM is able to reconstruct fine-scale, high-frequency features that IsoNet fails to maintain. The magnified insets highlight a critical difference: while CryoLVM correctly reconstructs continuous pore-like channels, IsoNet produces fragmented or occluded segments that result in topologically incorrect representations of these hollow structures. The FSC curves comparing reconstructed maps to reference models show that CryoLVM maintains a higher correlation across mid-to-high spatial frequencies and reaches established cutoff thresholds at higher spatial frequencies than IsoNet (Fig. 6), indicating better resolution restoration.

Table 3: Performance of different methods in missing wedge restoration task. The score is computed between two half maps (predicted map and ground truth map). Additional results are in Appendix G.6.

| Method | phenix.mtriage | |
| | FSC-0.143 ↓ | FSC-0.5 ↓ |
|---|---|---|
| IsoNet | 10.448 | 12.361 |
| CryoLVM | **10.094** | **11.447** |

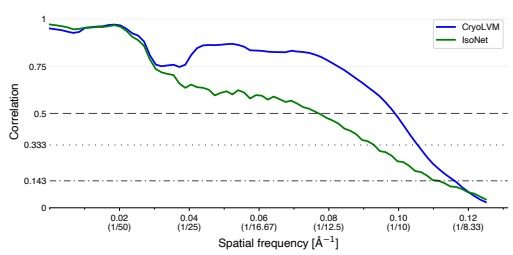

Figure 6: FSC score versus spatial frequency for predicted maps of IsoNet and CryoLVM (EMD-5331).

## 4.5 ABLATION & DISCUSSION

To validate our design choices and understand the contribution of individual components, we conducted ablation studies examining key architectural decisions, pretraining strategies, loss configurations, and postprocessing methods. In this section, we briefly present the key findings.

**SCUNet-based models achieve better performance than their ViT counterparts across all evaluated downstream tasks.** To demonstrate that adopting SCUNet as the backbone over ViT yields consistently better performance across all downstream tasks, we compared models with different backbones trained using identical hyperparameters; detailed results are presented in Appendix G.1.

**Composite loss of MSE and HistKL leads to accelerated convergence and enhanced downstream performance.** To validate the contribution of the proposed histogram KL loss, we compared fine-tuning CryoLVM on the density map super-resolution task with the composite loss against with MSE alone; extended results are provided in Appendix G.2.

**JEPA pretraining yields transferable representations that improve downstream performance.** To assess the effectiveness of our pretraining approach, we compared the pretrained-then-finetuned model against a model trained from scratch, and further compared JEPA against masked autoencoder (MAE) pretraining on the map sharpening task. The pretrained model demonstrates consistent improvements across all metrics, and JEPA consistently outperforms MAE; extended results are presented in Appendix G.3.

## 5 CONCLUSION

In this paper, we present CryoLVM, the first foundation model for cryo-EM density maps that employs Joint-Embedding Predictive Architecture with SCUNet-based backbone to learn rich structural representations. We introduce a novel histogram-based distribution alignment loss that accelerates convergence and enhances fine-tuning performance. CryoLVM consistently outperforms established baselines across three critical downstream tasks, showing robust performance on genuinely noisy, low-resolution experimental maps encountered in real-world cryo-EM workflows. We envision that this foundation model paradigm will facilitate broader adoption of unified AI-driven approaches in cryo-EM and contribute to accelerating structural biology discoveries.

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

## A    STATEMENT OF THE USE OF LARGE LANGUAGE MODELS

Large language models were used solely to polish the academic writing of this paper. They were not involved in research ideation, experimental design, data analysis, or any other substantive aspects of the work. The authors take full responsibility for the content of the manucript.

## B    ADDITIONAL RELATED WORK

### B.1    FOUNDATION MODELS FOR CRYO-EM

The emergence of foundation models has begun transforming the cryo-EM data processing landscape, with recent works addressing distinct stages of the cryo-EM structural determination pipeline. DRACO (Shen et al., 2024) introduced a denoising-reconstruction autoencoder pretrained over 270,000 cryo-EM movies or micrographs, demonstrating strong generalization capabilities across micrograph-level tasks including denoising, micrograph curation, and particle picking. CryoFastAR (Zhang et al., 2025) pioneered geometric foundation modeling for cryo-EM by directly predicting particle poses for unordered, noisy 2D projection images, facilitating acceleration in ab initio reconstruction compared to traditional iterative optimization approaches. Cryo-IEF (Yan et al., 2024) presented a comprehensive foundation model pretrained on approximately 65 million particle images and showed excellent performance in tasks such as classifying particles from different structures, clustering particles by pose, and assessing image quality. These works operate on 2D cryo-EM images from the data acquisition and reconstruction stages and complement CryoLVM to address complete cryo-EM workflow from particle image processing through density map post-processing and analysis.

### B.2    JEPA APPLICATIONS IN BIOLOGICAL CONTEXT

Joint-Embedding Predictive Architecture (JEPA), first introduced by Assran et al. (2023). with I-JEPA, has gained significant traction in computational biology due to its ability to learn semantic representations through prediction in abstract embedding space rather than raw pixel-level reconstruction. Brain-JEPA (Dong et al., 2024) provides the most direct precedent for CryoLVM by successfully applying JEPA to 3D spatiotemporal biological data using functional Magnetic Resonance Imaging (fMRI). This pioneering brain dynamics foundation model incorporates two innovative

techniques tailored to volumetric biological data: Brain Gradient Positioning (BGP), which establishes functional coordinate system for brain functional parcellation and enhances the positional encoding of different Regions of Interest (ROI), and Spatiotemporal Masking, which is designed for the unique characteristics of fMRI data to tackle with heterogeneous time-series patches. Brain-JEPA achieves state-of-the-art performance in dempgraphic prediction, disease diagnosis, and trait prediction, demonstrating JEPA's effectiveness for complex 3D biological volumes with temporal dependencies.

### B.3 SCUNet applications in image denoising

The SCUNet framework represents a breakthrough in hybrid architecture for image denoising, effectively combining global modeling capabilities of Swin Transformers (Liu et al., 2021) with local feature extraction advantages of convolutional neural networks. Zhang et al. (2023) introduced SCUNet in their seminal work, proposing Swin-Conv (blocks) as the core building components of a UNet backbone architecture, where each SC block processes input through a $1 \times 1$ convolution followed by evenly splitting feature maps into two groups that are respectively fed into Swin Transformer blocks and residual convolutional layers. Subsequent developments further validated SCUNet's effectiveness across diverse denoising applications: SUNet (Fan et al., 2022) applies Swin Transformer layers as basic building blocks within UNet architecture for image denoising, achieving significant performance improvements on fixed-size input denoising tasks, while SCNet (Lin et al., 2025) proposes a dual-branch fusion network that combines Swin Transformer branches with ConvNext (Liu et al., 2022b) branches, employing Feature Fusion Blocks with joint spatial and channel attention for adaptive output merging, demonstrating robustness under severe noise conditions and proving effective in real-world applications like mural image denoising.

## C Additional Details of CryoLVM

### C.1 Implementation Details

Fig. 7 illustrates the detailed architecture of CryoLVM's Context Encoder and Target Encoder. Both encoders share an identical hierarchical design, with Target Encoder parameters updated via exponential moving average (EMA) of Context Encoder during pretraining. The encoder architecture begins with an initial $3 \times 3 \times 3$ convolution applied to the input density, followed by three consecutive down-sampling stages. Each stage consists of a Swin-Conv Block and a $2 \times 2 \times 2$ convolution. Within each Swin-Conv Block, feature map is first passed through a $1 \times 1 \times 1$ convolution, and then split evenly into two feature map groups, each of which is then fed into a swin transformer (SwinT) block and a residual $3 \times 3 \times 3$ convolution (RConv) block separately; after that, feature maps are concatenated and passed through a final $1 \times 1 \times 1$ convolution to integrate local and global modeling.

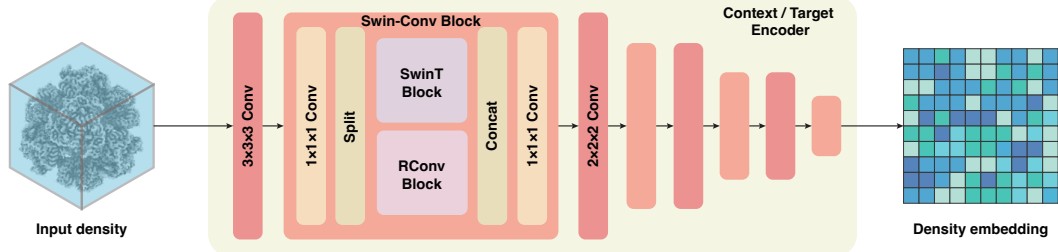

Figure 7: Detailed architecture of CryoLVM's Context Encoder and Target Encoder.

Fig. 8 illustrates the complete architecture of CryoLVM including a pretrained encoder and a task-specific decoder for downstream applications. The modular design enables effective transfer learning, where the encoder captures generalizable density map representations in the pretraining stage, while the decoder specializes these features for particular cryo-EM tasks. The task-specific decoder employs a symmetric up-sampling strategy to progressively reconstruct high-quality density maps from the encoded density embeddings. It starts with three consecutive up-sampling stages, each comprising a $2 \times 2 \times 2$ transposed convolution followed by a Swin-Conv Block. The transposed

convolution performs spatial up-sampling while reducing feature dimensionality, and the Swin-Conv Block refines the upsampled features. A critical component is the skip connections between corresponding encoder and decoder layers. Specifically, skip connections link the convolution layers of encoder with the transposed convolution layers of decoder. These connections preserve multi-scale structural information that might otherwise be lost. After that, feature map is passed through a final $3 \times 3 \times 3$ convolution to produce output density.

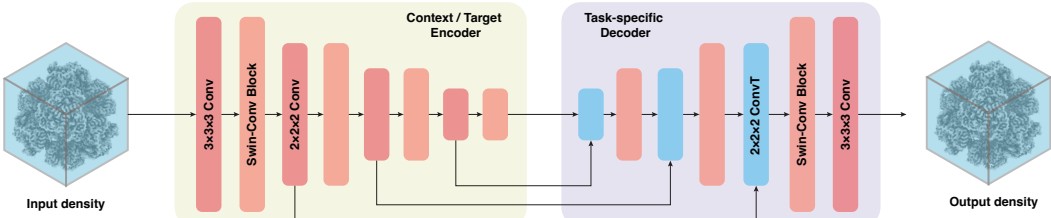

Figure 8: Detailed implementation of pretrained encoder and task-specific decoder for downstream application.

# D    DATA CURATION AND PROCESSING

## D.1    PREPROCESSING FOR TRAINING ACCELERATION AND MEMORY REDUCTION

**Pretraining**    To construct a high-quality pretraining dataset of cryo-EM density maps with high-quality structural information for CryoLVM, we focused on relatively high-resolution experimental density maps with corresponding resolved PDB(Berman et al., 2000) structures deposited in EMDB (wwPDB Consortium, 2023). We utilized Cryo2StructData, currently the most comprehensive publicly available depository in this domain, containing 7,600 cryo-EM density maps with associated structural annotations. We assembled our pretraining dataset from the training subset of Cryo2StructData, which comprises 7,392 density maps representing proteins and macromolecular complexes with resolutions spanning 1-4 Å. We first excluded density maps present in the test sets of baseline methods used for downstream task comparisons to guarantee meticulous evaluation and prevent data leakage, resulting in a final pretraining corpus of 7,302 density maps. Next, we preprocessed the pretraining density maps as follows: 1) voxel sizes were uniformly resampled to 1 Å spacing, 2) density values were clipped to the 0.01-0.99 percentile range to mitigate outlier effects and subsequently normalized to [0,1], 3) density maps are converted to Pytorch tensor format with bFloat16 precision, providing computational speedup and memory reduction.

**Density map sharpening**    We first constructed our test set by selecting 50 cryo-EM density maps from the established EMReady's benchmark, ensuring direct comparison with prior methods. To build our training and validation sets, we collected additional single-particle cryo-EM maps at 3-6 Å resolutions and their corresponding atomic structures from EMDB (wwPDB Consortium, 2023) and PDB (Berman et al., 2000). We applied stringent quality control criteria to filter map-model pairs: 1) do not contain backbone atoms only, 2) do not include unknown residues, 3) do not include missing chain, 4) have orthogonal map axis, 5) resolution is given by the FSC-0.143 threshold, 6) any chain in the model does not share $>30\%$ sequence identity with any chain in the test set models. This yielded a total of 400 pairs of map and model as the training set, and 70 pairs of map and model as the validation set. Target density maps were simulated from their corresponding PDB structures using Chimera.molmap (Pettersen et al., 2004) with the resolution parameter matched to that of the paired experimental maps. We preprocessed the paired maps as follows: 1) voxel sizes were uniformly resampled to 1 Å spacing, 2) density values were clipped to the 0.01-0.99 percentile range to mitigate outlier effects and subsequently normalized to [0,1], 3) density maps are converted to Pytorch tensor format with bFloat16 precision, providing computational speedup and memory reduction. During training, we employed a sliding window sampling strategy with stride 24 to generate $48^3$ input volumes and augmented training data through random flips and random rotations.

**Density map super-resolution**  We constructed a test set consisting of 40 single-particle cryo-EM density maps, a training set with 400 maps and a validation set with 50 maps. All density maps exhibit resolutions ranging from 2.3-6 Å and satisfy the quality control criteria described above for our density map sharpening experiment. Target density maps were simulated from their corresponding PDB structures using Chimera.molmap with the resolution parameter uniformly set to 1.8 Å. We preprocessed the paired maps following the same procedure from the density map sharpening experiment.

**Missing wedge restoration**  In the procedure for restoring a missing wedge, input density maps come from Cryo-ET, which typically exhibit resolutions worse than 8 Å, even after subtomogram averaging. Thus, we chose maps with resolutions lower than 8 Å and applied angular masks to mimic the missing wedge effect. Given constraints from the Nyquist sampling theorem and computational efficiency, a voxel size of 4 Å was used instead of 1 Å, which was used for the density map super-resolution and sharpening tasks. In total, we obtained 400 training samples, 50 validation samples, and 40 test samples. The resolution distributions of the training and test sets are shown in Fig. 9.

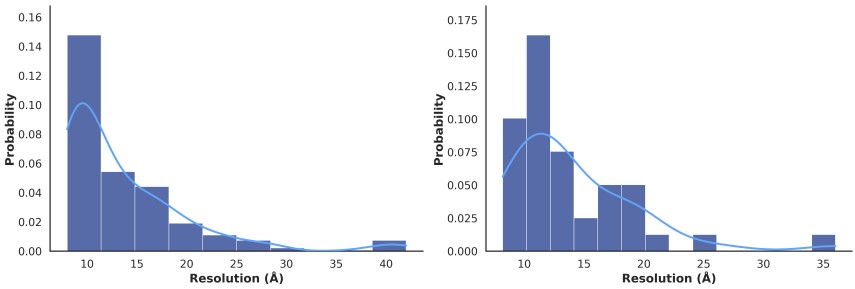

Figure 9: Resolution Distribution of Missing Wedge Restoration Task. (Left) Training Set. (Right) Test Set.

---

**Algorithm 1:** Gaussian-weighted fusion of patch predictions into a full density map

---

**Input:** Overlapping patch predictions $\{\hat{\mathbf{y}}_k\}_{k=1}^K$, their voxel coordinates $\{\mathbf{p}_k\}_{k=1}^K$, box size $B$, stride $s$, small $\varepsilon > 0$

**Output:** Reconstructed volume $\mathbf{M}$

**Require:** 3D Gaussian weight kernel $\mathbf{W}_{\text{ker}} \in \mathbb{R}^{B \times B \times B}$ with entries

$w_{i,j,\ell} = \exp\left(-\frac{\|(i,j,\ell)-\mathbf{c}\|_2^2}{2\sigma^2}\right)$, normalized to $[1,3]$ (center larger, border smaller); zero-initialized accumulators $\mathbf{V}$, $\mathbf{S}$ sized to the (padded) volume. (Implementation mirrors the sliding-window accumulation with per-voxel weights and final normalization.)

**for** $k = 1, 2, \ldots, K$ **do**
  Extract the region-to-update $\mathcal{R}_k \subset \mathbf{V}$ starting at $\mathbf{p}_k$ with size $B \times B \times B$
  // Weighted accumulation of prediction and weights
  $\mathbf{V}[\mathcal{R}_k] \leftarrow \mathbf{V}[\mathcal{R}_k] + \hat{\mathbf{y}}_k \odot \mathbf{W}_{\text{ker}}$
  $\mathbf{S}[\mathcal{R}_k] \leftarrow \mathbf{S}[\mathcal{R}_k] + \mathbf{W}_{\text{ker}}$
**end**
// Element-wise normalization to resolve overlaps
$\tilde{\mathbf{M}} \leftarrow \mathbf{V} \odot \max(\mathbf{S}, \varepsilon)^{-1}$
Crop $\tilde{\mathbf{M}}$ to the original (unpadded) spatial extent to obtain $\mathbf{M}$
**return** $\mathbf{M}$

---

### D.2 Postprocessing After Prediction

We performed post-processing on the predicted density maps to ensure consistency, reduce artifacts, and improve interpretability of reconstructed volumes. Given patch-based predictions, overlapping regions were fused utilizing a Gaussian-weighted scheme (See in Algo. 1), which balances contributions between central and boundary voxels to promote continuity. This blending step reduces discontinuities at patch boundaries while maintaining local structural details. After weighted fusion, we

applied normalization to maintain consistent voxel intensity distributions across the reconstructed volume. This step corrects for potential intensity variations arising from the overlapping patch-based prediction strategy, where edge regions of patches may exhibit different contrast characteristics than central regions due to reduced spatial context during inference. The final step involved cropping out padded regions introduced during the patch-wise sliding-window prediction step. Together, these steps generate a clean, continuous and interpretable density map suitable for model building, resolution evaluation, and quantitative validation. The Tab. 12 shows the post processing ablation study for missing wedge restoration task.

## E  EXPERIMENTAL SETTINGS

For downstream training, we used the AdamW optimizer with an initial learning rate of $3 \times 10^{-5}$, weight decay of $1 \times 10^{-3}$, and a ReduceLROnPlateau (factor 0.9, patience 5). All experiments were run for 500 epochs using a batch size of 32, mixed precision (bfloat16), and DDP (Distributed Data Parallel). Input density maps were cropped into $48^3$-voxel volumes, with a stride 24 during training and 48 during validation. To boost model robustness, we applied random flips along all three axes and random rotations as data augmentation. The backbone architecture was based on SCUNet, which takes a single input channel, uses a base dimension of 32 with head dimension 16, and consists of seven stages configured as [2,2,2,2,2,2,2]. Swin-Conv Blocks with a local attention window of $3^3$ and zero drop-path rate were used; the output was one-channel density map. This design strikes a balance between local convolutional features and global attention, enabling efficient training on large multi-GPU clusters while preserving structural detail. The specific hyperparameters for downstream task training are listed in Tab. 4 and further architectural details are availble in Tab. 5.

Table 4: Hyperparameters for downstream task training

| Parameter | Value | Description |
|---|---|---|
| optimizer | AdamW | Optimizer type |
| learning rate | $3 \times 10^{-5}$ | Initial LR for AdamW |
| weight_decay | $1 \times 10^{-3}$ | Weight decay coefficient |
| epochs | 500 | Number of training epochs |
| batch_size | 35 | Batch size per process |
| loss | $\alpha \mathcal{L}_{\text{MSE}} + (1 - \alpha) \mathcal{L}_{\text{HistKL}}, \alpha \in [0, 1]$ | Reconstruction loss ($\ell_2$) |
| lr_scheduler | ReduceLROnPlateau | Factor 0.9, patience 5 (min mode) |
| mixed precision | bfloat16 | `autocast` enabled |
| distributed | DDP (NCCL) | Multi-GPU data parallel training |
| input size | $48^3$ | Input/output volume size |
| train stride | 24 | Sliding-window stride (training) |
| val stride | 48 | Sliding-window stride (validation) |
| augmentation | RandomFlip/RandomRotate | 3D flips and $90°$ rotations |
| num_workers | 10 | DataLoader workers per process |
| pin_memory | True | Page-locked host memory |

## F  EVALUATION METRICS

**Fourier Shell Correlation (FSC).** The Fourier Shell Correlation (FSC) is a standard metric for measuring the resolution of reconstructed density maps in cryo-EM. It is computed between two independently reconstructed half-maps $F_1(\mathbf{k})$ and $F_2(\mathbf{k})$, as the normalized cross-correlation of their Fourier coefficients within a frequency shell $S$:

$$\text{FSC}(S) = \frac{\sum\limits_{\mathbf{k} \in S} F_1(\mathbf{k}) \, F_2^*(\mathbf{k})}{\sqrt{\sum\limits_{\mathbf{k} \in S} |F_1(\mathbf{k})|^2 \, \sum\limits_{\mathbf{k} \in S} |F_2(\mathbf{k})|^2}}, \tag{6}$$

Table 5: Model architecture of SCUNet

| Component | Setting | Description |
|---|---|---|
| backbone | SCUNet | Swin-Conv U-Net for volumetric data |
| input channels | 1 | `in_nc = 1` |
| stages / blocks | [2, 2, 2, 2, 2, 2, 2] | Blocks per stage (`config`) |
| base dim | 32 | Channel width at first stage (`dim`) |
| head_dim | 16 | Attention head dimension |
| window_size | 3 | Local attention window ($3\times3\times3$) |
| input_size | 48 | Input size for model |
| drop_path_rate | 0 | Stochastic depth disabled |
| activation | (inherited by SCUNet) | Follows SCUNet default |
| output | 1 channel | Reconstructed high-frequency density |

where $F_i(\mathbf{k})$ denotes the Fourier transform of the $i$-th half-map, $*$ indicates complex conjugation, and the summation is performed over all Fourier coefficients $\mathbf{k}$ within the shell $S$. The FSC curve typically decays with increasing frequency, and the resolution is conventionally defined at the spatial frequency where FSC drops below a certain threshold (e.g., 0.143 or 0.5).

**$d_{model}$** . The $d_{model}$ metric serves as an effective resolution indicator that quantifies the level of structural detail present in cryo-EM maps. It measures the effective resolution at which the atomic model agrees with the experimental density, providing a practical evaluation of density map's interpretability.

**Map–Model Correlation Coefficients (Phenix).** The Phenix validation suite provides several correlation coefficients (CCs) to quantify the agreement between an atomic model and a cryo-EM density map (Afonine et al., 2018):

- $CC_{box}$: correlation computed over the entire map volume;

- $CC_{mask}$: correlation restricted to voxels inside a molecular mask;

- $CC_{peaks}$: correlation evaluated at the strongest density peaks;

- $CC_{volume}$: correlation calculated for a user-specified molecular volume, typically representing the region occupied by the model.

These metrics probe complementary aspects of map-model agreement: global consistency, localized fit within the molecular envelope, and correspondence at prominent density features, and agreement within the molecular volume of interest respectively.

**Q-score** The Q-score quantifies atomic resolvability in cryo-EM maps (Pintilie et al., 2020). For each atom, the local density profile is extracted from the map and compared against a reference Gaussian profile, with the correlation the two serving as the Q-score. A higher Q-score indicates that the atomic density is well resolved and closely resembles the expected Gaussian profile, while lower values reflect poorer local resolvability. We report the average Q-score over all atoms in the model, as implemented in Chimera's MapQ plugin, to evaluate overall map quality.

**CryoRes** CryoRes (Dai et al., 2023) is a deep learning-based method for estimating local resolution from a single cryo-EM density map. Unlike traditional approaches, it does not require half-maps or multiple reconstructions, making it broadly applicable in scenarios where only a final map is available. CryoRes predicts voxel-wise resolution estimates that highlight spatial variability in map quality. These local resolution annotations provide complementary information to global metrics, offering a more fine-grained assessment of map reliability.

# G ADDITIONAL EXPERIMENT RESULTS

## G.1 ABLATION STUDY OF SCUNET-BASED BACKBONE VS VIT-BASED BACKBONE

The ablation results validate the effectiveness of SCUNet-based backbones relative to ViT backbones across density map sharpening, super-resolution, and missing wedge restoration. As shown in Tab. 6, 7, and 8, as well as in the radar plots, SCUNet consistently outperforms ViT on both correlation- and resolution-based metrics. For comparability, metrics where lower values indicate better performance (e.g., FSC-based metrics) were negated so that all scores follow a higher-is-better convention, consistent with $CC_{box}$ and related measures. The unified radar plot (Fig.10) summarizes these results, providing a holistic view of backbone performance across evaluation metrics.

Table 6: Evaluation metrics of SCUNet-based and ViT-based backbone models on density map sharpening task.

|  | phenix.map_model_cc | | | Chimera.MapQ |
| --- | --- | --- | --- | --- |
|  | $CC_{box}$ ↑ | $CC_{mask}$ ↑ | $CC_{peaks}$ ↑ | Q-score ↑ |
| CryoLVM (ViT) | 0.826 | 0.733 | 0.739 | 0.387 |
| CryoLVM (SCUNet) | **0.894** | **0.821** | **0.806** | **0.444** |

Table 7: Evaluation metrics of SCUNet-based and ViT-based backbone models on density map super-resolution task.

|  | phenix.mtriage | | | CryoRes |
| --- | --- | --- | --- | --- |
|  | $d_{model}$(Å) ↓ | FSC-0.143(Å) ↓ | FSC-0.5(Å) ↓ | Resolution(Å) ↓ |
| CryoLVM (ViT) | 2.35 | 2.65 | 4.73 | 3.41 |
| CryoLVM (SCUNet) | **2.33** | **2.58** | **4.58** | **3.39** |

Table 8: Evaluation metrics of SCUNet-based and ViT-based backbone models on missing wedge restoration task.

|  | phenix.mtriage | | phenix.map_model_cc | | |
| --- | --- | --- | --- | --- | --- |
|  | FSC-0.143(Å) ↓ | FSC-0.5(Å) ↓ | $CC_{box}$ ↑ | $CC_{mask}$ ↑ | $CC_{volume}$ ↑ |
| CryoLVM (ViT) | 16.51 | 22.15 | 0.355 | 0.277 | 0.241 |
| CryoLVM (SCUNet) | **10.09** | **11.45** | **0.391** | **0.391** | **0.348** |

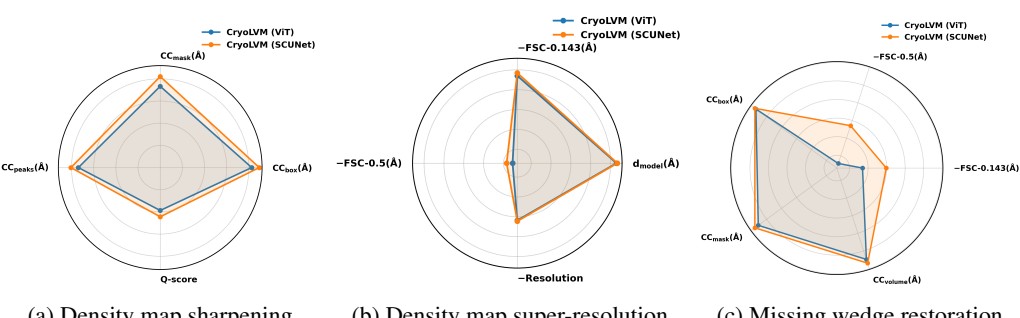

(a) Density map sharpening    (b) Density map super-resolution    (c) Missing wedge restoration

Figure 10: Radar plots comparing CryoLVM performance with SCUNet versus ViT backbones across three tasks.

## G.2 ABLATION STUDY OF COMPOSITE LOSS OF MSE AND HISTKL

The experimental results summarized in Tab. 9 demonstrate that the proposed HistKL loss, when used in conjunction with the standard MSE loss, consistently outperforms the baseline MSE-only approach across all evaluation metrics in the density map super-resolution task. Beyond the improvement of density map quality, the incorporation of HistKL loss also accelerates the training convergence. Our analysis reveals that the composite loss configuration reaches optimal validation loss at epoch 107, whereas the MSE-only baseline requires 279 epochs to achieve its best validation loss—representing a $2.6\times$ reduction in training time. These results highlight the effectiveness and efficiency of the HistKL term in guiding model optimization.

Table 9: Impact of different training losses on model performance.

| | phenix.mtriage | | | CryoRes |
|---|---|---|---|---|
| | Resolution(Å) $\downarrow$ | FSC-0.143(Å) $\downarrow$ | FSC-0.5(Å) $\downarrow$ | Resolution(Å) $\downarrow$ |
| CryoLVM (MSE) | 2.36 | 2.63 | 4.61 | 3.45 |
| CryoLVM (Composite) | **2.33** | **2.58** | **4.58** | **3.39** |

## G.3 ABLATION STUDY OF PRETRAINING

To validate the effectiveness of our JEPA-based pretraining approach, we conducted two complementary ablation studies on the density map sharpening task. First, we compared our pretrained-then-finetuned model against a model trained from scratch using identical architectures and hyperparameters (Table 10). The pretrained model demonstrates consistent improvements across all metrics. These results confirm that self-supervised pretraining on high-quality cryo-EM density maps enables the model to learn transferable structural representations that enhance downstream task performance. Second, we evaluated JEPA against the widely-used Masked-Autoencoder (MAE) pretraining approach (Table 11). While both methods benefit from pretraining, JEPA consistently outperforms MAE across all metrics.

Table 10: Evalutation metrics of pretrained-then-finetuned and trained-from-scratch models on density map sharpening task.

| | phenix.map_model_cc | | | Chimera.MapQ |
|---|---|---|---|---|
| | $CC_{box}$ $\uparrow$ | $CC_{mask}$ $\uparrow$ | $CC_{peaks}$ $\uparrow$ | Q-score $\uparrow$ |
| CryoLVM (Scratch) | 0.878 | 0.808 | 0.786 | 0.437 |
| CryoLVM (Pretrain) | **0.894** | **0.821** | **0.806** | **0.444** |

Table 11: Evalutation metrics of JEPA-pretrained and MAE-pretrained models on density map sharpening task.

| | phenix.map_model_cc | | | Chimera.MapQ |
|---|---|---|---|---|
| | $CC_{box}$ $\uparrow$ | $CC_{mask}$ $\uparrow$ | $CC_{peaks}$ $\uparrow$ | Q-score $\uparrow$ |
| CryoLVM (MAE) | 0.881 | 0.813 | 0.791 | 0.441 |
| CryoLVM (JEPA) | **0.894** | **0.821** | **0.806** | **0.444** |

## G.4 ABLATION STUDY OF POSTPROCESSING METHOD

During inference, CryoLVM processes density maps using a sliding window approach with overlapping patches, necessitating an effective fusion strategy to merge predictions from overlapping regions into a coherent output volume. We conducted an ablation study comparing two postprocessing methods: mean-weighted fusion, which assigns equal weight to all overlapping predictions, and Gaussian-weighted fusion, which assigns higher weights to central regions of each patch and lower

weights to boundary regions (as detailed in Alg. 1). To better differentiate between these two approaches and amplify their differences, we adopted a larger stride of 24 compared to the previously used value of 12.

As shown in Tab. 12, Gaussian-weighted fusion consistently outperforms mean-weighted fusion across all evaluation metrics for the missing wedge restoration task.

Table 12: Comparison of two postprocessing strategies

|  | phenix.mtriage | | phenix.map_model_cc | | |
|---|---|---|---|---|---|
|  | FSC-0.143(Å) ↓ | FSC-0.5(Å) ↓ | $CC_{box}$ ↑ | $CC_{mask}$ ↑ | $CC_{volume}$ ↑ |
| CryoLVM (Mean) | 10.66 | 13.07 | 0.390 | 0.388 | 0.345 |
| CryoLVM (Gaussian) | **9.92** | **11.47** | **0.391** | **0.391** | **0.348** |

### G.5 ADDITIONAL RESULTS OF SUPER RESOLUTION TASKS

This subsection presents extra results regarding the density map super-resolution task. As shown in Fig. 11, CryoLVM achieves state-of-the-art performance overall, outperforming all baselines in both the FSC-0.5 and the CryoRes metrics. Fig. 12 visualizes improvement in local resolution for the EMD-0023 example: the deposited map spans a local resolution range of 3.39 Å to 4.12 Å, whereas CryoLVM's predicted map compresses that range to a 2.31 Å to 3.45 Å. This illustrates that CryoLVM not only enhances global resolution measures but also recovers finer structural detail in regions where the original map is weaker.

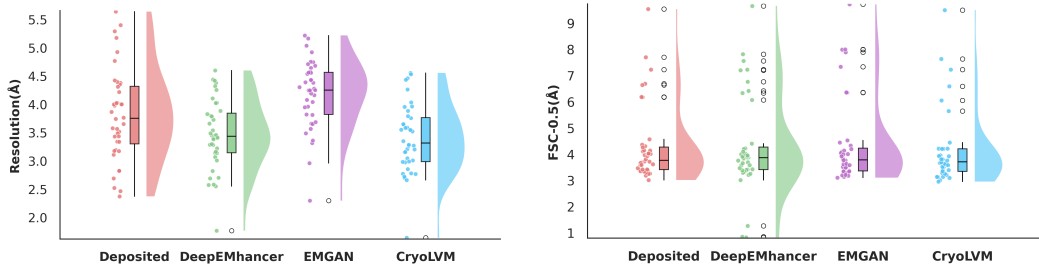

Figure 11: Comparison of CryoRes estimates and FSC-0.5 for density map super-resolution task.

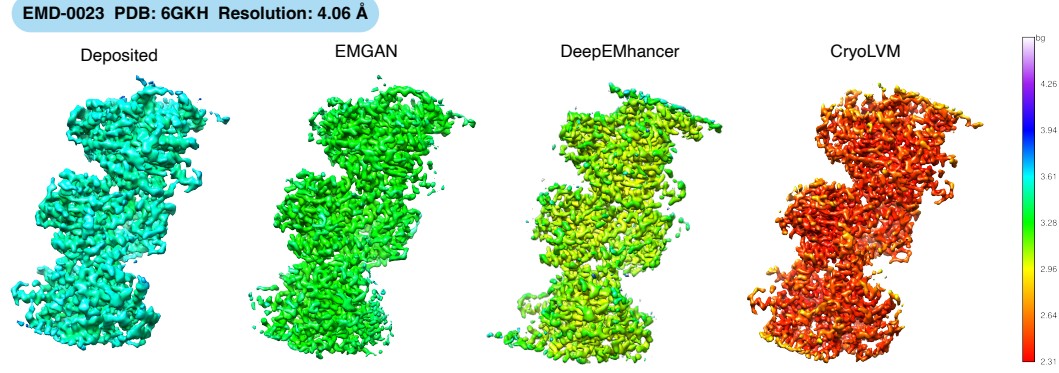

Figure 12: Comparison of local resolution maps between different methods on case EMD-0023. Local resolution maps are calculated via CryoRes and visualized through Chimera.

### G.6 ADDITIONAL RESULTS OF MISSING WEDGE RESTORATION

To provide thorough evaluation of missing wedge restoration performance, we present additional quantitative metrics and visualizations beyond the main results reported in Section 4.4. As shown

in Tab. 13, CryoLVM achieves higher correlation coefficients in three complementary assesment criteria: 1) $CC_{box}$, which measures global structural coherence by evaluating correlation across the entire reconstruction volume; 2) $CC_{mask}$, which quantifies reconstruction accuracy within a maksed region; and 3) $CC_{volume}$, which assesses volumetric fidelity through voxel-wise correlation analysis of the complete density distribution. These improvements indicate that CryoLVM not only recovers missing angular information more accurately but also maintains superior structural consistency. Fig. 14 presents the FSC-based resolution metrics (FSC-0.143 and FSC-0.5), where CryoLVM exhibits significantly better resolution distributions, indicating more effective recovery of high-frequency structural information lost due to missing wedge effects. Fig. 15 provides an additional visualization case for EMD-5106.

Table 13: Addition CC results for missing wedge restoration task. Metrics computed using phenix.map_model_cc.

| | phenix.map_model_cc | | |
| --- | --- | --- | --- |
| | $CC_{box} \uparrow$ | $CC_{mask} \uparrow$ | $CC_{volume} \uparrow$ |
| IsoNet | 0.364 | 0.371 | 0.328 |
| CryoLVM | **0.391** | **0.391** | **0.348** |

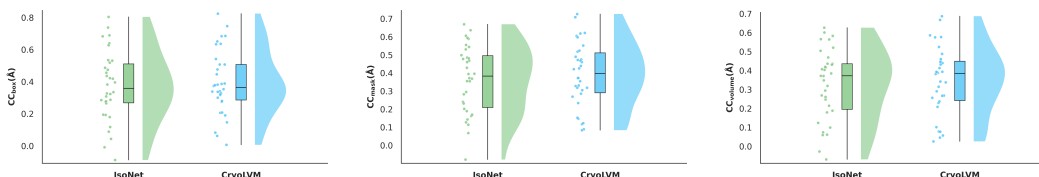

Figure 13: CC-based performance evaluation for missing wedge restoration task.

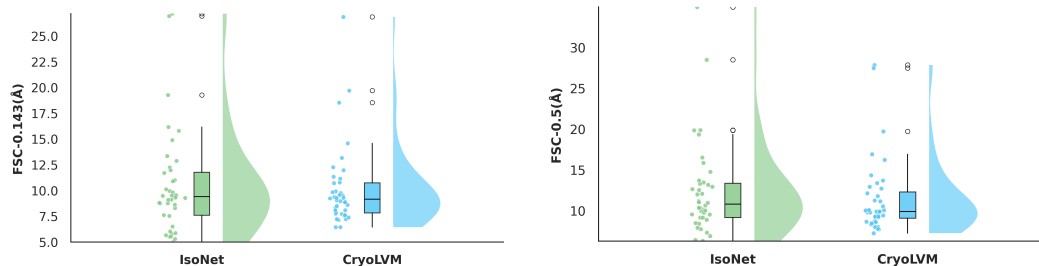

Figure 14: FSC-based performance evaluation for missing wedge restoration task.

## G.7    VALIDATION OF IMPACT ON AUTOMATED MODEL BUILDING

To demonstrate the practical impact of CryoLVM processing on downstream structural biology workflows, we conducted a case study using ModelAngelo, a state-of-the-art automated model building tool, on EMD-6656 (PDB: 5H30, resolution: 3.5 Å). As shown in Fig. 16, we compared atomic structures built from the deposited experimental map versus the CryoLVM-sharpened map. The CryoLVM-processed map yielded quantifiable improvements in model quality: RMSD to the reference structure decreased from 0.68 Å to 0.58 Å, and sequence match increased from 92% to 94.3%. These gains show that enhanced map quality directly translates to more accurate and complete automated model building, reducing manual intervention requirements in structural determination workflows.

## G.8    PROTEIN SECONDARY STRUCTURE CLASSIFICATION

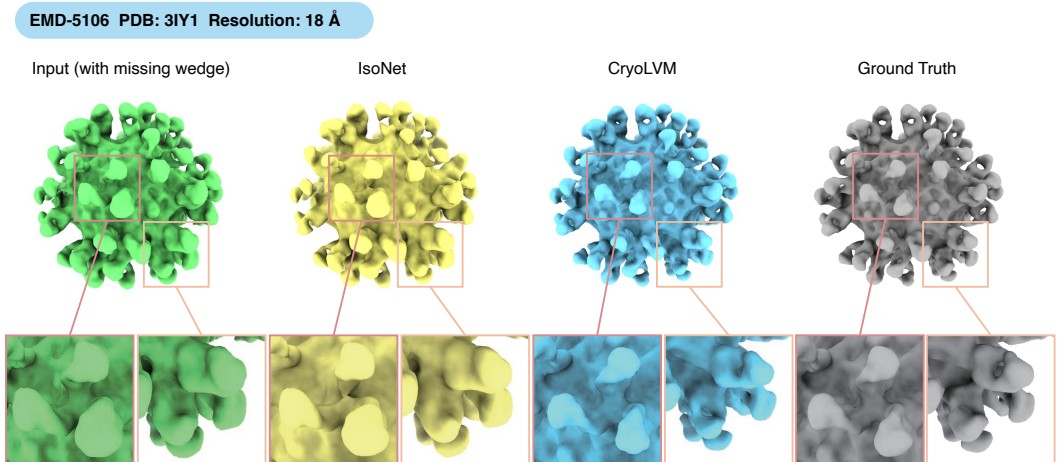

Figure 15: Additional visualization case for missing wedge task. The original density map is EMD-5106 from EMDB and is added wedge-shaped mask for model processing.

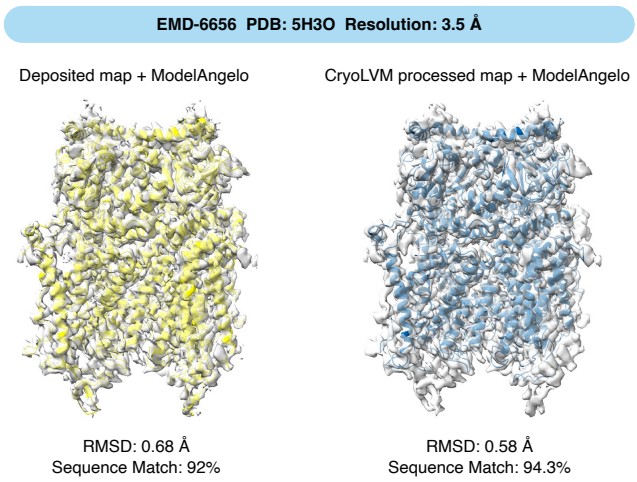

Figure 16: Comparative model building quality assessment using ModelAngelo. Atomic structures were built automatically from (Left) the deposited experimental map EMD-6656 (resolution: 3.5 Å) and (Right) the same map after CryoLVM sharpening.

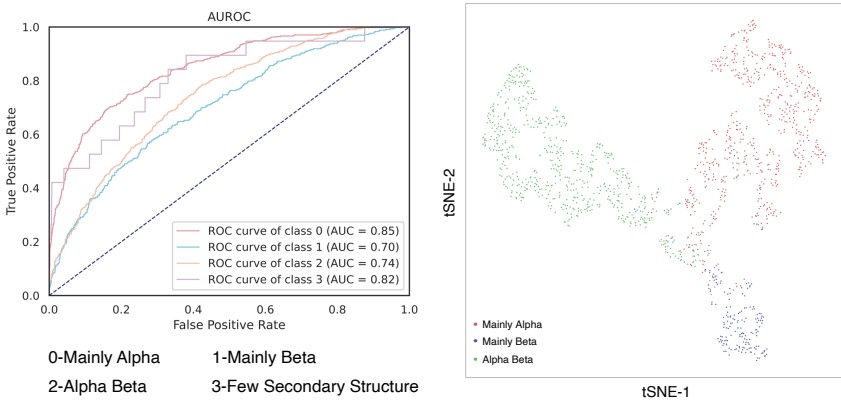

Figure 17: Protein secondary structure classification and representation visualization results.

