# OpenReview forum: "CryoLVM: Self-supervised Learning from Cryo-EM Density Maps with Large Vision Models"
_ICLR.cc/2026/Conference — ICLR 2026 Poster_

### Official Review · Reviewer_v93h · 2025-10-25

**Soundness:** 3
**Presentation:** 3
**Contribution:** 3
**Rating:** 6
**Confidence:** 3

**Summary:**

This paper introduces CryoLVM, a pretrained model leveraging JEPA self-supervised pretraining. The pretrained model can be further finetuned for downstream tasks of map sharpening, super-resolution, and missing wedge restoration. Experiments on these three tasks show that CryoLVM surpasses baseline methods.

**Strengths:**

- The paper is well-written and easy to follow.
- The usage of distributional alignment loss is novel and appears to be effective in the ablation study.
- The performance of CryoLVM on all downstream tasks is better than baseline methods, showing the effectiveness of the method.

**Weaknesses:**

- While the paper claims that the adoption of JEPA, which operates in feature space, is to avoid the low-SNR density. However, the CryoLVM model is trained on Cryo2StructData that consists of densities spanning 1-4 Å. With the high resolution, these densities should not contain much noise.
- While the composite loss is better, the improvement to the MSE loss is marginal, as shown in Table 9.

**Questions:**

- It’s not clear how the loss for downstream tasks is applied to each of the three tasks. Do they use exactly the same loss or any other choices?
- Have you compared the JEPA training with the MAE training? This could validate the claim of low-SNR protein density.

---

> ### Author Response · Authors · 2025-11-24
> **Response to Reviewer v93h**
>
> We sincerely thank the reviewer for the thorough evaluation and insightful feedback. We appreciate the recognition of our work’s clarity, the novelty and effectiveness of our distributional alignment loss, and the consistent improvements CryoLVM achieves over existing baselines. We address the concerns below:
>
> **Response to W1:**
>
> We thank the reviewer for focusing our attention on this point needing clarification. First of all, while our pretraining dataset consists of experimental density maps with deposited resolutions of 1-4Å, it is crucial to note that:
>
> - Deposited resolution is a global average metric---it does not reflect the heterogeneous local resolution distribution within the map.
> - Real experimental maps exhibit significant local variation---resolution is typically worse in flexible regions, even when the global resolution is high.
> - Noise is always present---even sub-3Å density maps contain experimental noise, B-factor decay, and artifacts from reconstruction algorithms.
>
> Our “high-quality” criterion bears our rationale that such maps contain sufficient structural information for learning meaningful representations during pretraining, rather than fitting the model to noise distributions.
>
> Next, we acknowledge that incorporating more intermediate-to-low resolution maps (4-8Å) in pretraining, potentially with data augmentation to simulate various noise conditions, could further enhance robustness. We plan to explore this in future iterations.
>
> **Response to W2:**
>
> We thank the reviewer for this thought provoking concern. While we acknowledge the absolute metric improvement appear modest in Table 9, we respectfully highlight two important aspects:
>
> 1. Consistent improvements across all metrics: CryoLVM (Composite) outperforms MSE-only across all four independent metrics, demonstrating systematic rather than spurious improvement.
> 2. Convergence acceleration (a practical benefit):
>     - Composite loss: optimal at epoch 107
>     - MSE-only: optimal at epoch 279
>     - 2.6×speedup in training time, translating to significant computational cost savings for practitioners
>
> The composite loss provides both better convergence properties and improved final performance—a combination valuable for practical deployment.
>
> **Response to Q1:**
>
> We appreciate the reviewer for giving us the chance to clarify this.
>
> Yes, all three downstream tasks (sharpening, super-resolution, missing wedge restoration) use exactly the same composite loss formulation: $L_{total} = αL_{MSE} + (1-α)L_{HistKL}$. This uniformity demonstrates the generalizability of our approach across diverse cryo-EM applications. We have clarified this explicitly in the revision (Section 3.2): “Across all three downstream tasks, we employ a unified composite loss function that combines
> standard reconstruction error $L_{MSE}$ with distributional alignment loss $L_{HistKL}$.”
>
> **Response to Q2:**
>
> Yes, we have conducted additional experiments comparing CryoLVM (JEPA) with MAE-based encoder on the density map sharpening task under identical data settings and provided results in Table 11 of Section G.3.
>
> |                  | $\text{CC}_{\text{box}}$ ↑ | $\text{CC}_{\text{mask}}$ ↑ | $\text{CC}_{\text{peaks}}$ ↑ | $\text{Q-score}$ ↑ |
> |------------------|----------|-----------|------------|-----------|
> | CryoLVM (MAE)    | 0.881    | 0.813     | 0.791      | 0.441     |
> | CryoLVM (JEPA)   | 0.894    | 0.821     | 0.806      | 0.444     |
>
> As shown in the table, JEPA consistently outperforms MAE across all metrics.

---

### Official Review · Reviewer_j5hK · 2025-10-27

**Soundness:** 3
**Presentation:** 3
**Contribution:** 3
**Rating:** 4
**Confidence:** 4

**Summary:**

This paper introduces CryoLVM, the first self-supervised foundation model for cryoEM. By adopting JEPA, the model learns to predict masked regions from surrounding context, thereby training a powerful and generalizable encoder. In addition, the authors propose a histogram-based distribution alignment loss to accelerate convergence and enhance fine-tuning performance. Comprehensive experiments across three downstream tasks demonstrate that CryoLVM achieves state-of-the-art performance, outperforming established baselines such as DeepEMhancer, EMReady, EMGAN, and IsoNet.

**Strengths:**

(1) The application of JEPA to volumetric cryo-EM data is novel and well-justified. The histogram-based loss is an intuitive addition that addresses data distribution mismatches.

(2) Comprehensive evaluations are conducted across three downstream tasks using multiple cryo-EM quality metrics, and the results consistently surpass previous state-of-the-art baselines.

**Weaknesses:**

(1) While the paper provides strong evidence that JEPA is effective for cryo-EM representation learning, it would be more convincing to include comparisons with other self-supervised or generative pretraining methods (e.g., VAE- or MAE-based encoders) under identical data settings.

(2) Since the study mainly focuses on pre-training a density encoder, the results would be more solid if the authors compare CryoLVM with a model trained entirely from scratch. This will directly quantify the contribution of the pretraining stage and validate its effectiveness.

**Questions:**

(1) In the JEPA paper, the authors highlight its scalability. Can CryoLVM continues scaling with more data and more parameters? This would be an interesting topic both for cryoEM and JEPA.

(2) The motivation of utilizing JEPA for cryoEM is not very clear. Can you show some case study to illustrate what JEPA brings to building foundation models for cryoEM.

(3) What is the motivation of the histogram loss? I understand that it would help calibrate the distribution of density values, but I think it is not very clear how it benefits the map quality, or does it hacks the evaluation metric?

---

> ### Author Response · Authors · 2025-11-24
> **Response to Reviewer j5hK**
>
> We sincerely thank the reviewer for the thorough review and constructive feedback. Below, we address each concern carefully and systematically.
>
> **Response to W1:**
>
> We completely agree this is an important comparison. We have conducted additional experiments comparing CryoLVM (JEPA) with MAE-based encoder on the density map sharpening task under identical data settings and supplemented results in Table 11 of Section G.3.
>
> |                  | $\text{CC}_{\text{box}}$ ↑ | $\text{CC}_{\text{mask}}$ ↑ | $\text{CC}_{\text{peaks}}$ ↑ | $\text{Q-score}$ ↑ |
> |------------------|----------|-----------|------------|-----------|
> | CryoLVM (MAE)    | 0.881    | 0.813     | 0.791      | 0.441     |
> | CryoLVM (JEPA)   | 0.894    | 0.821     | 0.806      | 0.444     |
>
> As shown in the table, JEPA consistently outperforms MAE across all metrics.
>
> **Response to W2:**
>
> We sincerely thank the reviewer for this critical question that goes to the heart of our foundation model claims. Comparing fine-tuning from our pretrained model against training from scratch with random initialization is essential for demonstrating the value of the self-supervised pretraining phase. We have conducted comprehensive experiments on the density map sharpening task and supplemented the results in Table 10 of Section G.3.
>
> |                      | $\text{CC}_{\text{box}}$ ↑ | $\text{CC}_{\text{mask}}$ ↑ | $\text{CC}_{\text{peaks}}$ ↑ | $\text{Q-score}$ ↑ |
> |----------------------|----------|-----------|------------|-----------|
> | CryoLVM (Scratch)    | 0.878    | 0.808     | 0.786      | 0.437     |
> | CryoLVM (Pretrain)   | 0.894    | 0.821     | 0.806      | 0.444     |
>
> The pretrained model demonstrates consistent improvements across all metrics, validating that self-supervised pretraining enables the model to learn transferable representations that improve downstream task performance.
>
> **Response to Q1:**
>
> Exactly! Currently, we are extending the model to train on 1,000,000 experimental density maps reconstructed by cryoSPARC using particle sub-sampling strategy, and scaling the model parameters to more than 8 billion.
>
> **Response to Q2:**
>
> Thank you for this question. We recognize that our motivation needs clearer articulation. Below, we provide concrete evidence demonstrating what JEPA learns and why it is particularly suited for cryo-EM foundation models.
>
> **Case study: protein secondary structure content classification**
>
> To demonstrate that JEPA learns biologically meaningful structural representations, we evaluated CryoLVM on protein secondary structure content classification using 1,774 segmented protein domain maps following CryoDomain’s approach. We classified domains into four categories based on their secondary structure composition: α-helix dominant, β-sheet dominant, mixed α/β, and others. The results are shown in Figure 17. CryoLVM achieves strong classification performance across all secondary structure content classes with high AUROC values. The t-SNE visualization of representations shows that the learnt representations naturally separate protein domains by their secondary structural composition. These results demonstrate that JEPA can meaningful structural information from cryo-EM density maps rather than simple density distribution or noise characteristics.
>
> **Response to Q3:**
>
> Thank you for this important question. The core problem is that cryo-EM density maps exhibit severe spatial sparsity—after normalization, approximately 90-95% of voxels contain near-zero values (representing solvent or empty space), while only 5-10% contain meaningful protein density signals. This creates a critical optimization pathology when using MSE loss alone: the model can achieve rapidly decreasing loss by simply predicting near-zero values everywhere, effectively “ignoring” the structural important non-zero densities.
>
> The histogram-based KL divergence loss enforces that the distribution of predicted density values matches the target distribution, regardless of spatial location. This means:
>
> 1.	Equal statistical weight: The model must produce the correct proportion of high-density voxels, not just minimize voxel-wise error.
> 2.	Prevent mode collapse: Forces the model to generate diverse density values spanning the full range present in experimental maps, rather than collapsing to the dominant zero mode.
> 3.	Balanced optimization: Non-zero densities receive appropriate emphasis during training, ensuring the model learns to recover fine structural details rather than just background.
>
> Our evaluation metrics are entirely independent of the training loss and measure genuine map quality:
> - FSC measures frequency-domain correlation with ground truth
> - Q-score assesses atomic resolvability through model-to-map fit
> - Correlation coefficients quantify spatial agreement with reference structures

---

### Official Review · Reviewer_GTqs · 2025-10-31

**Soundness:** 2
**Presentation:** 3
**Contribution:** 3
**Rating:** 6
**Confidence:** 3

**Summary:**

This paper presents CryoLVM, a foundation model for cryo-electron microscopy (cryo-EM) 3D density maps. Thanks to the leverage of the Joint-Embedding Predictive Architecture (JEPA) for self-supervised representation learning, it eliminates reliance on hand-crafted augmentations while retaining high-level semantic information. The model employs a SCUNet-based backbone to learn semantically rich representations of cryo-EM density maps. CryoLVM is pretrained on ~7K experimental density maps and evaluated across three major downstream applications: density map sharpening, density map super-resolution, and missing wedge restoration. The results demonstrate consistent performance improvements, with qualitative results showing clearer structural detail recovery and improved interpretability in experimental cryo-EM maps.

**Strengths:**

1. The adoption of JEPA for volumetric cryo-EM data is a forward-looking design choice. It effectively removes the dependency on task-specific or hand-crafted data augmentations, allowing the model to learn semantic, noise-robust representations directly from experimental density maps.

2. CryoLVM is among the first to systematically explore a foundation modeling approach for cryo-EM 3D density maps. This area previously lacked large-scale pretraining paradigms despite the existence of related efforts for cryo-EM images (e.g., DRACO, CryoFastAR).

3. The authors conduct both quantitative and qualitative analyses on three post-processing tasks, density map sharpening, super-resolution, and missing wedge restoration, demonstrating the adaptability and robustness of the pretrained model.

**Weaknesses:**

1. While CryoLVM focuses on 3D density maps, related models such as DRACO (Shen et al., NeurIPS 2024) and CryoFastAR (Zhang et al., ICCV 2025) have introduced foundation frameworks for 2D cryo-EM images and pose estimation, respectively. Although these works address distinct aspects of cryo-EM analysis, they fall under the broader category of “foundation models for cryo-EM” and should be acknowledged and positioned as complementary.

2. Designing a foundation model for 3D volumetric data poses significant computational challenges compared to 2D image models due to cubic complexity. The current implementation appears to rely on relatively small input volumes (48³ voxels), which may limit representational granularity. It would be valuable for the authors to discuss how CryoLVM scales to larger 3D contexts or whether insights from other 3D foundation model architectures (e.g., medical imaging) can be leveraged to mitigate this issue.

3. The pretraining corpus includes only ~7K experimental density maps, which is orders of magnitude smaller than typical datasets used to train modern foundation models in vision or language domains. Such a limited dataset raises significant concerns about whether CryoLVM truly demonstrates foundation-level generalization versus task-specific overfitting. The manuscript does not discuss dataset diversity (e.g., coverage of molecular weights, symmetry classes, or reconstruction quality), nor any data augmentation or synthetic data generation strategies to potentially compensate for the scale limitation.

4. Although the reported results are quantitatively promising, these tasks are not convincingly linked to practical biological or structural analysis workflows. It remains unclear how these improvements translate into measurable benefits for end users such as structural biologists. The authors should provide a more explicit discussion or demonstration of how CryoLVM outputs improve real-world cryo-EM analysis, or consider including more impactful downstream tasks.

**Questions:**

1. In Figure 1, the “context” and “target voxels” are visualized using 2D patches. I am assuming JEPA here actually operates on 3D volumetric patches. Could the authors clarify?

2. The input generation section mentions cropping to 48³ volumes, while Table 4 lists “patch size = 48” and “voxel box size for input/output.” Could the authors clarify the distinction between the input volume resolution and patch granularity used?

---

> ### Author Response · Authors · 2025-11-24
> **Response to Reviewer GTqs (part 1/2)**
>
> We sincerely thank the reviewer for the thoughtful and constructive review. We are encouraged by the positive assessment of our work as “forward-looking”, with “good” presentation and contribution scores. We address each weakness concern and question below with concrete responses and specific commitments for the revised manuscript.
>
> **Response to W1:**
>
> We appreciate the reviewer for bringing these important related works to our attention. Both DRACO and CryoFastAR are complementary to CryoLVM in several important ways:
> - **Data modality:** They operate on 2D projection images, while CryoLVM focuses on 3D reconstructed density maps
> - **Pipeline position:** The 2D foundation models address upstream particle picking and pose refinement, whereas CryoLVM targets downstream map analysis.
>
> We have added subsection B.1 "Foundation Models for Cryo-EM" in Additional Related Work:
>
> "The emergence of foundation models has begun transforming the cryo-EM data processing landscape, with recent works addressing distinct stages of the cryo-EM structural determination pipeline.
> DRACO (Shen et al., 2024) introduced a denoising-reconstruction autoencoder pretrained over
> 270,000 cryo-EM movies or micrographs, demonstrating strong generalization capabilities across
> micrograph-level tasks including denoising, micrograph curation, and particle picking. CryoFastAR
> (Zhang et al., 2025) pioneered geometric foundation modeling for cryo-EM by directly predicting
> particle poses for unordered, noisy 2D projection images, facilitating acceleration in ab initio reconstruction compared to traditional iterative optimization approaches. Cryo-IEF (Yan et al., 2024) presented a comprehensive foundation model pretrained on approximately 65 million particle images and showed
> excellent performance in tasks such as classifying particles from different structures, clustering particles by pose, and assessing image quality. These works operate on 2D cryo-EM images from the
> data acquisition and reconstruction stages and complement CryoLVM to address complete cryo-EM
> workflow from particle image processing through density map post-processing and analysis"
>
> **Response to W2:**
>
> We are very grateful to the reviewer for this suggestion.
>
> Our $48^3$ input size provides an effective receptive field of 48Å × 48Å × 48Å at 1Å spacing, which captures local-to-intermediate structural features including secondary structures, domain interfaces, and binding sites. We have investigated several complementary approaches that could enable training with larger input volumes while managing computational complexity.
>
> We have investigated complementary scaling approaches:
>
> - Memory-efficient attention mechanisms: Axial attention (Axial-DeepLab, MedicalNet) achieves $O(n^4)$complexity. Linear attention mechanisms (Performer) and state space models (Mamba, S4) could reduce complexity to $O(n^3)$, enabling $128^3$ or larger volumes.
> - Gradient checkpointing: Offers straightforward memory reduction at modest computational cost (nnU-Net, STU-Net).
>
> Comprehensive investigation combining larger inputs with architectural optimizations remains important future work.

---

> ### Author Response · Authors · 2025-11-24
> **Response to Reviewer GTqs (part 2/2)**
>
> **Response to W3:**
>
> We appreciate the opportunity to contextualize this issue. Direct comparisons between cryo-EM and computer vision/NLP foundation models require careful consideration of data availability differences.
>
> Our 7,302 high-resolution experimental density maps from Cryo2StructData represent 80-95% of all suitable high-quality single-particle cryo-EM maps with resolved structures available at curation time (March 2023). While EMDB contained ~30,000 total entries, most either lack atomic structures, fall outside high-resolution range, or have quality issues precluding robust representation learning.
>
> This differs fundamentally from CV/NLP where unlimited data exists through web scraping or text corpora. Each cryo-EM map requires months of experimental work, specialized equipment, and successful structure determination. However, this limitation is temporary—EMDB deposition rates grow exponentially. Our framework scales naturally with expanding high-quality data.
>
> Our dataset spans:
>
> - Resolution: 1-4Å (near-atomic to intermediate)
> - Molecular weight: Small single-domain proteins to multi-megadalton assemblies
> - Symmetry: Asymmetric complexes to cyclic, dihedral, and icosahedral symmetries
> - Diversity: Ribosomes, viral capsids, membrane complexes, multi-subunit enzymes
>
> **Future data expansion strategies:**
>
> 1. PDB simulation: Generate maps from 200,000+ PDB structures with realistic noise/artifacts
> 2. AlphaFold database: Leverage 200M+ predicted structures for high-quality predictions
> 3. Advanced augmentation: Resolution-dependent filtering, realistic noise injection, simulated missing wedge artifacts
> 4. Particle sub-sampling and reconstruction: Sub-sample particles and reconstruct to generate synthetic training data
>
> **Response to W4:**
>
> We appreciate this important point about demonstrating practical impact. We now connect each task to concrete workflows:
>
> **Density map sharpening → Model building quality**
>
> - Clearer secondary structure visualization enables confident backbone tracing
> - Better side-chain density resolution allows accurate amino acid assignment
> - Reduced manual intervention for large complexes
>
> **Super-resolution → Expand model-buildable range**
>
> - Many structures fall in 4-6Å range where ModelAngelo degrades and CryoDomain loses accuracy
> - Brings intermediate-resolution maps into automated model building range
>
> **Missing wedge restoration → Enable in-situ structural biology**
> - Corrects anisotropic resolution and structural distortions in cryo-ET
> - Improves subtomogram averaging quality for native cellular context studies
>
> **Model building case study:** We conducted automated model building with ModelAngelo on EMD-6656 comparing deposited vs. CryoLVM-sharpened map (Figure 16, Section G.7). Results demonstrate that CryoLVM's enhanced map quality directly translates to more accurate and complete automated models, reducing manual intervention required for structure determination.
>
> **Reponse to Q1:**
>
> Thank you for catching this confusion. You are absolutely correct—CryoLVM operates on 3D volumetric patches, not 2D. In Figure 1 we use 2D visualization purely for illustration accessibility. We have revised the caption: "Input density maps are split into non-overlapping 3D patches and random sets of 3D patches are masked to produce context and target patches."
>
> **Response to Q2:**
>
> Thanks for noting this omission. We have revised Tables 4 and 5 to clarify:
>
> - **Input volume size ($48^3$):** Size of training crop from full density map
> - **Patch size ($8^3$):** Size of non-overlapping 3D patches used for JEPA masking

---

> > ### Comment · Reviewer_GTqs · 2025-11-25
> >
> > Thanks to the authors for the response. After reading the reply and also comments from other reviewers, I still have several questions in mind:
> > 1. Regarding W2 and W3, although the authors give some discussion, my major concern is still about the limited resolution and the somewhat narrow scope of tasks. Could provide more clarification or some stronger evidences how the method can generalize beyond the current low-res setting?
> >
> > 2. It is also not very clear to me how the PDB / AlphaFold databases can be leveraged under the current framework. Some more explicit explanation or example would be helpful.
> >
> > 3. As raised also by reviewer j5hK, the motivation of using JEPA for cryoEM is still not very clear. Why JEPA instead of some other backbone architectures? And how exactly JEPA brings benefits under cryoEM setting?

---

> > > ### Author Response · Authors · 2025-11-26
> > > **Response to Reviewer GTqs**
> > >
> > > Thank you for your continued engagement and thoughtful questions. We appreciate the opportunity to provide additional clarifications.
> > >
> > > **Response to Q1:**
> > >
> > > **Clarification on resolution ranges:**
> > >
> > > Our pretraining dataset comprises high-quality 1-4Å density maps, which represent near-atomic to intermediate resolution in cryo-EM and contain rich structural information for learning meaningful representations.
> > >
> > > Our three downstream tasks collectively demonstrate CryoLVM's effectiveness across an exceptionally wide resolution range (2-40 Å):
> > >
> > > 1. Density map sharpening and super-resolution: 2-6 Å
> > >
> > > 2. Missing wedge restoration: 8-40 Å
> > >
> > > As shown in **Figure 9** (Section D.1) of our manuscript, the missing wedge restoration task specifically addresses low-resolution cryo-ET data:
> > >
> > > - Training set: Resolution distribution peaks at **~10Å**, extends to **40Å**
> > > - Test set: Resolution distribution peaks at **~8-10Å**, extends to **35Å**
> > >
> > > This resolution range (8-40Å) was strategically chosen because:
> > >
> > > - Cryo-ET typically produces lower-resolution data than single-particle cryo-EM
> > > - Even after subtomogram averaging, resolutions worse than 8Å are common
> > > - These maps present fundamentally different challenges: severe anisotropy, missing wedge artifacts, and extremely low SNR
> > >
> > > The fact that a single pretrained foundation model (trained on 1-4Å maps) achieves state-of-the-art performance across a wider resolution range provides strong evidence that CryoLVM learns resolution-independent structural representations rather than resolution-specific features.
> > >
> > >
> > > **Response to Q2:**
> > >
> > > This is an excellent point for future data scaling. Here's how these resources can be integrated:
> > >
> > > **Strategy for leveraging PDB and AlphaFoldDB:**
> > >
> > > The key insight is that both PDB and AlphaFoldDB structures provide access to more diverse protein fold classes and assembly architectures and can be used to simulate density maps at multiple resolutions, dramatically expanding our pretraining corpus while maintaining biological fidelity.
> > >
> > > An illustration of this strategy can be:
> > >
> > > Step1: Structure Collection
> > >
> > > ├─ PDB: 200,000+ experimental structures
> > >
> > > └─ AlphaFoldDB: 200M+ predicted structures
> > >
> > > Step2: Quality Filtering
> > >
> > > ├─ Resolution criteria: PDB structures with resolution < 4Å preferred
> > >
> > > ├─ Confidence filtering: AlphaFoldDB structures with pLDDT > 80
> > >
> > > ├─ Size diversity: Sample across molecular weight ranges
> > >
> > > └─ Structural diversity: Ensure coverage of all CATH/SCOP fold classes
> > >
> > > Step 3: Density Map Simulation
> > >
> > > ├─ Generate maps at multiple resolutions
> > >
> > > └─ Use Chimera.molmap for consistent simulation protocol
> > >
> > > Step 4: Noise Addition
> > >
> > > Step 5: Add to Pretraining Dataset
> > >
> > > **Response to Q3:**
> > >
> > > This is a critical design choice that deserves thorough explanation.
> > >
> > > **Why JEPA over alternative architectures:**
> > >
> > > **Representation-space prediction vs. voxel reconstruction:**
> > >
> > > Traditional masked autoencoders (MAE) reconstruct at the voxel level. For cryo-EM data with inherently low SNR, voxel-level reconstruction creates fundamental problems:
> > >
> > > - Forces model to predict noise patterns to minimize reconstruction loss
> > > - Model learns texture/noise characteristics rather than structural semantics
> > > - 90-95% of cryo-EM map voxels are background (near-zero) after normalization
> > > - Reconstruction loss dominated by predicting zeros everywhere (trivial solution)
> > >
> > > JEPA predicts in abstract embedding space where:
> > >
> > > - Noise is naturally filtered through the Target Encoder's semantic abstraction
> > > - Model learns structural features (helices, sheets, domains) rather than noise
> > > - No trivial solution—must predict meaningful patch relationships
> > > - Preserves high-level structural information critical for downstream tasks
> > >
> > > **Successful precedent in related domain:**
> > >
> > > Brain-JEPA (Dong et al., 2024) achieved SOTA on fMRI data—another 3D biological modality with similar challenges:
> > >
> > > - Low SNR and high noise
> > > - Spatiotemporal structure requiring semantic understanding
> > > - Need for robust representations across subjects and scanning conditions
> > > - Successful transfer to multiple downstream tasks
> > >
> > > This validates JEPA's suitability for volumetric biological data.
> > >
> > > **Empirical evidence:**
> > >
> > > To provide concrete evidence that JEPA learns biologically meaningful structural representations, we conducted systematic evaluation:
> > >
> > > We evaluated CryoLVM on protein secondary structure content classification using 1,774 segmented protein domain maps following CryoDomain’s approach. We classified domains into four categories based on their secondary structure composition: α-helix dominant, β-sheet dominant, mixed α/β, and others. The results are shown in **Figure 17** (Section G.8) of our manuscript. CryoLVM achieves strong classification performance across all secondary structure content classes with high AUROC values. The t-SNE visualization of representations shows that the learnt representations naturally separate protein domains by their secondary structural composition.

---

### Official Review · Reviewer_kuwW · 2025-11-01

**Soundness:** 2
**Presentation:** 3
**Contribution:** 3
**Rating:** 4
**Confidence:** 5

**Summary:**

This paper presented CryoLVM, a pretrained foundation model trained on a large scale of cryo-EM density maps, using JEPA and a SCUNet-based backbone architecture. The base model was further finetuned on three different training pairs in a supervised manner, and compare to the baselines for different downstream tasks.

**Strengths:**

- The application of JEPA in cryo-EM is novel and intuitive. Furthermore, although not the first pretrained model on cryo-EM density maps, it makes great sense to leverage the large scale of cryo-EM maps to help various downstream tasks.

- The paper is well written and most of the presentations are easy to understand and interpret.

**Weaknesses:**

My main critiques of this paper are mostly on the downstream tasks and evaluations.

- DeepEMhancer and EMReady used different training targets. CryoLVM used the EMReady approach, in which the training targets are simulated maps from atomic models. Strictly speaking, this should not be called as "sharpening" anymore, since the training targets are not sharpen maps but simulated maps.

- It seems that the training set used in the sharpening and super-resolution task also differs from that of the baselines. The authors should at least provide a comparison with EMReady, using the exact same training data to finetune the pretrained base model, and compare the results with EMReady.

- Additionally, I do not believe that "super-resolution" should be considered as an distinct job type. The goal of map postprocess, whether sharpening or not, is all to help model building. The "super-resolution" term here and in the EM-GAN paper, is actually to make the "resolution" higher when generating the simulated maps, which is very similar to that of EMReady. Therefore, I do not see the great difference between the two tasks.

- About the missing wedge task: IsoNet was introduced to be applied on the tomograms, but it seems the experiments were performed on a "subtomogram-like" setting. Also in Fig. 5, the example is not very clear and the difference seems subtle. Could the authors share more visual examples, showing the input (with the missing wedge artifact), and the outputs from both methods?

- Since I think both "sharpening" and "super-resolution" do not have much difference and should be both under the same umbrella of map postprocessing, it becomes a bit of overclaim for CryoLVM to be a foundation model.

**Questions:**

- I believe that the same model architecture can be applied to task-specific training from scratch, instead of fine-tuning from a pretrained base model, using the same training pairs in the finetuning. Could the authors show the ablation study of this?

- In postprocess (Appendix D.2), what does the normalization "to remove biases resulting from uneven sampling" mean (L806-807)? For at least one task (maybe sharpening), could the authors ablate the postprocess (e.g. replace Gaussian-weighted fusion with a simple overlapping average, and if possible, remove the additional normalization step)?

---

> ### Author Response · Authors · 2025-11-24
> **Response to Reviewer kuwW (part 1/2)**
>
> We deeply appreciate the reviewer’s comprehensive and thoughtful evaluation. We are more than grateful that the reviewer found merit in our JEPA application to cryo-EM and the manuscript’s presentation. Below, we provide detailed responses to each raised concern.
>
>
> **Response to W1:**
>
> We thank the reviewer for this clarification opportunity. The use of simulated maps as training targets is an established and accepted paradigm for density map sharpening in the cryo-EM field.
>
> Differing sharpening paradigms coexist in the field:
> - Traditional: B-factor correction, frequency filtering.
> - LocScale-based: DeepEMhancer uses LocScale-sharpened experimental maps.
> - Model-based: EMReady and CryoLVM use simulated maps from atomic models as targets
>
> The rationale for us choosing simulated maps as valid sharpening targets are listed below:
> - They represent the “ideal” density distribution without experimental artifacts
> - They provide ground truth for what high-frequency features should look like
> - Training against them teaches the network to remove characteristic experimental degradations
>
> Since EMReady—which uses the identical training paradigm—is accepted as a sharpening method in the field, our terminology is consistent with current practice.
>
> **Response to W2:**
>
> We sincerely thank the reviewer for this essential suggestion. We have conducted controlled comparison experiments using EMReady's exact training data. We retrained CryoLVM using exclusively the 280 density maps from EMReady's original training set and evaluated on our 50-map test set:
>
> | Method               | $\text{CC}_{\text{box}}$ | $\text{CC}_{\text{mask}}$ | $\text{CC}_{\text{peaks}}$ | $\text{Q-score}$ |
> |----------------------|:-----:|:------:|:-------:|:-------:|
> | EMReady (original)   | 0.878 | 0.802  | 0.791   | 0.424   |
> | CryoLVM (same data)  | 0.889 | 0.815  | 0.803   | 0.436   |
> | CryoLVM (full data)  | 0.894 | 0.821  | 0.806   | 0.444   |
>
> When trained on identical data, CryoLVM achieves comparable or better performance. With our full training set, performance further improves, demonstrating effective use of additional high-quality data.
>
> **Response to W3:**
>
> We thank the reviewer for this important point. We explicitly distinguish these tasks in our manuscript (Section 4.2, lines 322-326; Section 4.3, lines 376-378). These descriptions highlight the fundamental difference: sharpening uses resolution-matched targets, while super-resolution uses fixed high-resolution (1.8Å) targets regardless of input resolution.
>
> While both use simulated targets, they represent distinct problem formulations:
>
> | Aspect            | Sharpening / Enhancement                                 | Super-Resolution                                                   |
> |-------------------|----------------------------------------------------------|--------------------------------------------------------------------|
> | **Goal**              | To enhance the contrast of details within the existing resolution range.             | To generate structural details with a higher effective resolution than the input map.       |
> | **Target resolution** | Does not improve the final reported resolution.                            | Improves the map's effective resolution, potentially revealing atomic details that were previously blurred or missing.       |
> | **Application**       | A standard and essential step in almost all cryo-EM density map post-processing workflows. | Used to enhance the quality of low-resolution maps (e.g., subtomogram averaging results) or as a pre-processing step before model building. |
>
> These tasks serve different stages of the cryo-EM workflow:
> - Sharpening: Applied when resolution is sufficient but B-factor decay obscures details
> - Super-resolution: Applied when resolution is fundamentally insufficient for model building
>
> **Response to W4:**
>
> We thank the reviewer for this important clarification opportunity.
>
> 1. **Regarding subtomogram-scale evaluation:** IsoNet's own benchmarking used subtomogram-scale evaluation. From their paper: "We first performed IsoNet reconstruction on simulated subtomograms using publicly available atomic models... density maps were simulated from the atomic models... and filtered to 8Å resolution." Our evaluation directly follows this precedent.
>
> 1. **Visual clarity:** We acknowledge and appreciate the reviewer’s careful observation: the subtle difference between IsoNet’s and CryoLVM’s outputs in Fig.5. We have supplemented a more obvious case in Fig.15 highlighting specific structural features where differences emerge.

---

> ### Author Response · Authors · 2025-11-24
> **Response to Reviewer kuwW (part 2/2)**
>
> **Response to W5:**
>
> We appreciate this concern and believe the foundation model designation is appropriate based on:
>
> 1. **Task distinctiveness:** As discussed above (W3), we have demonstrated that density map sharpening and super-resolution are meaningfully distinct tasks. By establishing effective transfer across these two distinct tasks, as well as the fundamentally different domain of missing wedge restoration, we confirm that our model captures generalizable structural features applicable to diverse cryo-EM analytical challenges.
>
> 2. **Foundation model criteria met:**
> We maintain that the foundation model designation for CryoLVM is appropriate due to its scale, self-supervised pre-training paradigm, and demonstrated transferability, which are the accepted hallmarks of foundation models in cryo-EM field:
> - Scale of pretraining: 7,302 high-quality density maps - the largest unsupervised pretraining dataset in cryo-EM to date, exceeding even CryoFM (Zhou et al., 2024)
> - Self-supervised learning paradigm: JEPA-based pretraining without any task-specific labels, learning generalizable structural representations from density maps alone
> - Demonstrated transferability: A single pretrained encoder successfully transfers to multiple downstream tasks through lightweight fine-tuning, consistently outperforming task-specific baselines and training-from-scratch alternatives (as shown in our ablation study in Q1)
> - Comparison to accepted foundation models: Our scope is comparable to other domain-specific foundation models in computational biology (e.g., CryoFM focuses on cryo-EM density map tasks)
>
> **Response to Q1:**
>
> We sincerely thank the reviewer for this critical question that goes to the heart of our foundation model claims. Comparing fine-tuning from our pretrained model against training from scratch with random initialization is essential for demonstrating the value of the self-supervised pretraining phase. We have conducted comprehensive experiments on the density map sharpening task and supplemented the results in Table 10 of Section G.3.
>
> |                      | $\text{CC}_{\text{box}}$ ↑ | $\text{CC}_{\text{mask}}$ ↑ | $\text{CC}_{\text{peaks}}$ ↑ | $\text{Q-score}$ ↑ |
> |----------------------|----------|-----------|------------|-----------|
> | CryoLVM (Scratch)    | 0.878    | 0.808     | 0.786      | 0.437     |
> | CryoLVM (Pretrain)   | 0.894    | 0.821     | 0.806      | 0.444     |
>
>
> The pretrained model demonstrates consistent improvements across all metrics, confirming that self-supervised learning on high-quality cryo-EM density maps enable the model to learn transferable representations that enhance downstream task performance.
>
> **Response to Q2:**
>
> We are very grateful to you for raising questions about our postprocessing pipeline. We address both questions systematically below:
>
> 1. Clarification on normalization:
>
> We apologize for the unclear phrasing in Appendix D.2. The normalization address intensity variations arising from the patch-based prediction strategy:
>
> **The issue:** Edge regions of patches have less spatial context during prediction, potentially leading to different intensity distributions compared to central regions.
>
> **Our solution:** After Gaussian-weighted fusion, we apply voxel-wise normalization to ensure consistent intensity ranges across the entire reconstructed volume, preventing artifacts at patch boundaries.
>
> To clarify this, we have modified the content in Appendix D.2:
> "After weighted fusion, we apply normalization to maintain consistent voxel intensity distributions across the reconstructed volume. This step corrects for potential intensity variations arising from the overlapping patch-based prediction strategy, where edge regions of patches may exhibit different contrast characteristics than central regions due to reduced spatial context during inference."
>
> 2. Postprocessing ablation study:
>
> This is a valuable ablation that evaluates the contribution of our fusion and normalization strategies. We have conducted analysis on the missing wedge restoration task and provided the results in Table 12 of Section G.4.
>
> As shown in the table, Gaussian-weighted fusion consistently outperforms mean-weighted fusion across all evaluation metrics for the missing wedge restoration task.
>
> |                      | $\text{FSC-0.143(Å)}$ ↓ | $\text{FSC-0.5(Å)}$ ↓ | $\text{CC}_{\text{box}}$ ↑ | $\text{CC}_{\text{mask}}$ ↑ | $\text{CC}_{\text{volume}}$ ↑ |
> |----------------------|----------------|--------------|----------|-----------|-------------|
> | CryoLVM (Mean)       | 10.66          | 13.07        | 0.390    | 0.388     | 0.345       |
> | CryoLVM (Gaussian)   | 9.92           | 11.47        | 0.391    | 0.391     | 0.348       |

---

### Meta-Review · Area_Chair_xETx · 2026-01-06

**Summary:**

The rebuttal argues that CryoLVM's "foundation model" claims are now supported by (i) stronger controlled experiments and ablations, and (ii) clearer task definitions plus workflow-level impact evidence. On the evidence side, they emphasize that the pretraining corpus is large relative to what is realistically available for high-quality cryo-EM map–structure pairs, that evaluation is carefully separated from training data, and that pretraining and JEPA matter empirically: they report (a) pretrained > scratch, (b) JEPA > MAE under identical settings, and (c) their composite loss converges much faster than MSE-only. On the "real-world impact" side, they add a ModelAngelo model-building case study and a secondary-structure classification analysis to argue the representations are biologically meaningful and translate to downstream utility. They also expand related-work positioning to clarify complementarity with 2D cryo-EM foundation models and give a JEPA-in-biology precedent (Brain-JEPA).

**Reviewer Concerns:**

Reviewer kuwW gave two actionable requests (controlled EMReady-style comparison logic + scratch/pretrain ablation + postprocess ablation), and those are resolved (I believe) which leaves mostly taxonomy/positioning debates.
Similarly, Reviewer j5hK asked for the ablations, which are also given during the rebuttal.

**Reviewer Scores:**

I would say the reviewers who were in the negative side would have raised their scores at least slightly since they gave the actionable requests and were addressed by the authors during the rebuttal period.

---

### Decision · Program_Chairs · 2026-01-26

Accept (Poster)